# Microglial Senescence and Activation in Healthy Aging and Alzheimer’s Disease: Systematic Review and Neuropathological Scoring

**DOI:** 10.3390/cells12242824

**Published:** 2023-12-12

**Authors:** Antonio Malvaso, Alberto Gatti, Giulia Negro, Chiara Calatozzolo, Valentina Medici, Tino Emanuele Poloni

**Affiliations:** 1IRCCS “C. Mondino” Foundation, National Neurological Institute, Department of Brain and Behavioral Sciences, University of Pavia, 27100 Pavia, Italy; antonio.malvaso01@universitadipavia.it (A.M.); alberto.gatti07@universitadipavia.it (A.G.); 2Department of Neurology, University of Milano Bicocca, 20126 Milan, Italy; g.negro4@campus.unimib.it; 3Department of Neurology and Neuropathology, Golgi-Cenci Foundation, Abbiategrasso, 20081 Milan, Italy; c.calatozzolo@golgicenci.it; 4Department of Translational Medicine, University of Eastern Piedmont, 28100 Novara, Italy; valentina.v.medici@gmail.com

**Keywords:** human microglia, microglial senescence, microglial activation, Alzheimer’s disease, healthy aging, neuropathology

## Abstract

The greatest risk factor for neurodegeneration is the aging of the multiple cell types of human CNS, among which microglia are important because they are the “sentinels” of internal and external perturbations and have long lifespans. We aim to emphasize microglial signatures in physiologic brain aging and Alzheimer’s disease (AD). A systematic literature search of all published articles about microglial senescence in human healthy aging and AD was performed, searching for PubMed and Scopus online databases. Among 1947 articles screened, a total of 289 articles were assessed for full-text eligibility. Microglial transcriptomic, phenotypic, and neuropathological profiles were analyzed comprising healthy aging and AD. Our review highlights that studies on animal models only partially clarify what happens in humans. Human and mice microglia are hugely heterogeneous. Like a two-sided coin, microglia can be protective or harmful, depending on the context. Brain health depends upon a balance between the actions and reactions of microglia maintaining brain homeostasis in cooperation with other cell types (especially astrocytes and oligodendrocytes). During aging, accumulating oxidative stress and mitochondrial dysfunction weaken microglia leading to dystrophic/senescent, otherwise over-reactive, phenotype-enhancing neurodegenerative phenomena. Microglia are crucial for managing Aβ, pTAU, and damaged synapses, being pivotal in AD pathogenesis.

## 1. Introduction and Historical Hints

Microglial cells are central nervous system (CNS) resident macrophages that play important roles in development, homeostasis, and response to damage, infection, and microenvironment perturbations. Microglia differ from other types of brain cells because of their origin from the yolk sac instead from the neural tube [1,2,3]. Once penetrated into the nervous tissue, microglial cells can replicate and replenish into the brain, forming the population resident in the CNS [3,4]. With other glial cells, the microglia play important functions in brain homeostasis and regeneration [5], and microglial cells are considered to be a heterogeneous cell population within the CNS [6,7,8]. Indeed, healthy human microglial cells are an extraordinarily varied population inside the brain and play a myriad of supportive roles including cellular sensing, communication, degradation, and repair, with a large and significant transcriptomic profile revealed by different studies [3,9]. In addition to being the cornerstone of innate immunity and the first line of defense of the CNS from pathogens and internal injuries, microglia are involved in myelination, synapse remodeling, and tissue repair. Microglia and other CNS cells are in a continuous cross-talk with each other, thanks to soluble mediators (as growth factors, neurotransmitters, cytokines, chemokines, innate-immunity mediators, and tissue damage molecules) and extracellular vesicles containing active biomolecules that can modulate gene expression in distant cells [10]. Microglia and astrocytes are part, with the blood–brain barrier, of the so called “neuro-vascular unit” and are involved in the response to brain insults orchestrating neuroinflammation processes [11]. Moreover, microglia cooperate with oligodendrocytes in regulating the complex process of myelination under physiological and pathological conditions [12]. Microglial cells have been differentiated based on their morphology, numerousness, electrophysiological properties, and expression of markers [13,14,15], such as CD68, Human Leukocyte Antigen—DR isotype (HLA-DR), ionized calcium binding adaptor molecule 1 (Iba1), and arginase 1 (Arg1) [16,17,18], identified by traditional immunohistochemical techniques [19,20].

Due to their fundamental role in CNS defense and immunity, microglia promote inflammation and present the antigens to lymphocytes [21], but may also have anti-inflammatory properties. In different animal species, these actions are performed in different contexts and ways, and there are significant differences between murine microglia (frequently used as a study model) and human microglia [3,22]. To overcome the limits of using rodent models, recently microglia obtained by differentiation of human-induced pluripotent stem cells (iPSCs) have been employed in monocultures, 2D co-cultures, 3D organoids, and even transplanted into the mouse brain [23]. On the other hand, it is clearly necessary to study microglia directly in human brain tissue. Historically, microglial pathological alterations in the human postmortem brain were described by McGeer’s group in pioneering investigations about Alzheimer’s disease (AD) neuropathology [24]. Age-related microglia alterations with a moderately active phenotype were later reported [25]. Subsequent studies confirmed that microglial changes may be present in aged brain tissue from healthy controls [26,27]. During the last decades, microglia have been linked to the pathologic processes that underpin aging-related neurological dysfunction and illness. There is growing evidence that glial cells undergo an aging process, which may influence neurodegenerative disease progression [28,29,30].

The occidental population is already experiencing unparalleled long-life expectancy, implying that the prevalence of age-related disorders, such as neurological diseases, will increase. As a result, there is an urgent need to better understand the aging process in order to discriminate against the factors that cause neurodegenerative diseases. The comprehension of aging at a systemic level is difficult to reach, meaning that the best current strategy is to study aging at cellular level [31]. Regarding the brain, this entails knowing the impact of aging on neurons and glial cells and their respective interactions at the molecular level [32]. Aging in the human brain is of particular interest since age-related alterations are typically irreversible and progressively impair normal activity [33]. In the aging brain there is a gradual change in cellular environment that is the single most important risk factor for AD [34]. It is well known that aged glial cells can be classified as senescent or dystrophic [35]. Senescent microglia may lose supportive activities protecting neurons and mimic reactive microglia, secreting potentially noxious cytokines [36]. It is still uncertain if the two phenotypes “senescent” and “dystrophic” refer to the same state or are separate ones. Angelova et al. reviewed the differences between these two terms. The term “cellular senescence”, which originated in the research of cancer cells, denoted a loss of the ability to divide. In recent years, as suggested by several studies, this term has also been interpreted to mean the age-related shift in the secretory profile (senescence-associated secretory phenotype—SASP) [37,38,39]. Therefore, the term senescence can refer to a variety of functional modifications [39,40]. On the other hand, the definition of “dystrophic cells” implies the morphological change related to senescence, as observed by Streit et al. [26]. De-ramification and retraction of processes, the formation of aberrant swellings in residual processes, and cytoplasmic fragmentation or cytorrhexis could be considered morphological features of microglial dystrophy [3,27].

The terms “resting” and “active” microglia first appeared in the literature in the mid-1970s, after that Rio-Hortega recognized the remarkable morphological alteration of microglia after brain trauma [8,41,42,43]. More recently, to assist the study of microglial activation, the M1/M2 paradigm was conceived with M1 having a pro-inflammatory action and M2 producing anti-inflammatory effects [18,44,45,46,47,48,49]. This terminology dates back to the early 2000s, when immunologists identified the activation state of macrophages in vitro. They observed the classical “M1” pro-inflammatory pattern that is considered neurotoxic and the “M2” anti-inflammatory pattern that is considered neuroprotective [50]. The M1-M2 polarization of microglia corresponds to different active phenotypes implicated in detrimental inflammatory effects or in repair tasks [49,51,52]. Additionally, the term “M0′’ describes an in vitro non-active microglial state when cells are in culture with the presence of transforming growth factor β (TGFβ) and colony-stimulating factor-1 (CSF1). This peculiar phenotype probably represents a physiological homeostatic state of the microglial cells that are involved in functions other than M1 and M2, such as the CNS development [53]. Moreover, this classification is extensively used to express the idea that microglia can be harmful (M1) or beneficial (M2) depending on the microenvironment and pathophysiological conditions [18] (Figure 1). However, mounting evidence is integrating this simple distinction implying that microglial cells do not fall completely into the M1-M2 classification. Indeed, the M1-M2 phenotypes represent the two extremes of a wide range of possible intermediate cellular states, which in vivo perform numerous functions that cannot be reproduced in vitro [46].

This systematic review looks at the various phenotypes of human microglia, particularly reactive and senescent microglia, and their function in healthy aging and AD that is paradigmatic of age-related neurodegeneration. The interaction between the different microglial phenotypes, beta-amyloid, TAU protein, and iron metabolism will be discussed. We propose an integrated model of genetic, phenotypical, and neuropathological signatures of microglial dysfunction related to neurodegeneration. Furthermore, in light of the current evidence, we highlight potential strategies to study microglial senescence (MS) and we propose a semiquantitative score system to grade microglial activation.

## 2. Materials and Methods

### 2.1. Literature Search, Data Sources, and Studies Selection

We performed a systematic literature search on PubMed and Scopus up to June 2023 mentioning MS and microglial activation (MA) and different phenotypes in aging healthy controls and AD, using the terms: “Microglial Senescence”, “Microglial Activation” combined with “Human” OR “Humans” AND “Alzheimer Disease” OR “Alzheimer’s disease” OR “Healthy Aging” OR “Healthy Controls”. Additional articles were identified from other sources (i.e., articles cited in reviews). Those articles were imported to the PICO portal (automation tool software version 3.0.2023.1205) and duplicates were removed. The Appendix A and Figure 1 include search strategies with MeSH terms, inclusion and exclusion criteria, and PRISMA checklist reported according to the PRISMA statement 2020 (Preferred Reporting Items for Systematic Reviews and Meta-Analyses) guidelines [54,55]. The eligibility for inclusion was based on human MS and MA studies, reporting genetics, phenotypical, and neuropathological tissue signatures in healthy aging and AD. Concurrently, studies that were not relevant to human MS or MA or that presented limited data on the brain tissue samples, poorly developed methodologies, biased outcomes, and insufficient sample size were all ruled out.

The retrieved papers’ abstracts and titles were screened (first screening), and suitable articles were picked based on eligibility criteria. The full texts of selected papers were then screened again (second screening), and the articles that met the inclusion criteria were included in this review. Two investigators (AM and AG) performed the search and collected the data independently for internal validity, and the differences between the two researchers were compared, discussed, and the agreement was reached accordingly. The two investigators’ assessments were then cross-checked, and disagreements were resolved by a third reviewer (TEP) in order to reach a final decision. We registered the protocol on the International Platform of Registered Systematic Review and Meta-analysis Protocols (registration number: INPLASY2023110064).

### 2.2. Data Analysis, Visualization, and Data Availability

A systematic review of all included research was created. Concerns about the validity and imprecision of the results were addressed with overall judgments. Figures 1, 3, and 4 and the graphical abstract have been assembled using dynamic BioRender assets (icons, lines, shapes, and/or text). The final data set is available in Zenodo (https://doi.org/10.5281/zenodo.10154404). Any competent investigator may request access to anonymized data extracted from studies included in this publication.

## 3. Results

### 3.1. Literature Search

We found 2987 articles through the PubMed and Scopus search databases, after removing 62 duplicates and 978 records excluded for other reasons (e.g., no full-text available, articles not written in the English language, review articles, opinion articles, or book chapters). We excluded a total of 1658 articles after title/abstract screening, and no reports were retrieved. From the 289 articles assessed for eligibility, 242 articles were excluded for the following reasons: those that did not comprise Population, Intervention, Outcome, and Study Type (PICOs) statement and those that did not analyze intermediate microglial phenotypes in AD with respect to healthy aging. A final number of 47 articles were selected and included in the qualitative synthesis of our review (Figure 2; Table 1a,b shown at the end of the text).

### 3.2. Data Extraction, Synthesis, and Quality Assessment

Two reviewers (AM and AG) independently extracted essential information from each study, with discrepancies resolved by consensus. Data extracted included information of authors, population and methodology of the study, brain tissue conditions, whether it was comparable to other studies, study findings, and significant statistical value. A qualitative synthesis of data was employed to summarize the information obtained from the selected articles. Data were summarized using descriptive statistics, with means and standard deviations for continuous variables and frequencies and percentages for dichotomous variables. A meta-analysis (or quantitative analysis) was not performed due to inadequate uniform results from the articles chosen.

Two reviewers (AM and AG) independently assessed study quality and any discrepancies were resolved through discussion and with the expert opinion of a third reviewer (TEP). Quality of studies included in this systematic review was independently assessed by the two reviewers using the NIH Quality Assessment Tool for Case-Control studies/Systematic Reviews and Meta-Analysis (Study Quality Assessment Tools | NHLBI, NIH, https://www.nhlbi.nih.gov/health-topics/study-quality-assessment-tools (accessed on 20 October 2023) (Appendix A).

### 3.3. Population of Included Studies

In the scientific literature on the topic there is a large number of studies carried out on mouse models, while, overall, studies carried out on humans are relatively few. A total of 3630 cases were included in the qualitative analysis. Out of 3630 cases, 1503(41.30%) were Healthy Controls with male/female ratio of 0.49 and a mean age of 79.28 (±15.37 SD). Regarding the Healthy Controls group, 154 out of 1503 cases (10.25%) with a male/female ratio 0.20 and a mean age of 40 (±17.63 SD) were Younger Controls. The remaining parts were considered Older Controls (89.75%) with a male/female ratio 0.53 and a mean age of 79.30 (±17.54 SD).

Moreover, 1618 out of 3630 cases (44.57%) had Alzheimer’s disease with male/female ratio of 0.49 and a mean age of 79 (±15.07 SD). Out of 3630 cases, 449 (12.36%) had Mild Cognitive Impairment (MCI) with male/female ratio of 1.43 and a mean age of 82 (±13 SD). Out of 3630 cases, 49 (1.34%) were classified as non-AD dementia with male/female ratio of 0.48 and a mean age of 78.69 (±14.66 SD). Out of 3630 cases, 11 (0.30%) were classified as Lewy Bodies Dementia (LBD) with male/female ratio of 4.5 and a mean age of 81 years old.

## 4. Discussion

### 4.1. Human Microglial “Senescence” and “Dystrophy” in Normal Aging and AD

The term “senescence” embraces a wide spectrum of phenomena as diverse as microglia themselves. These phenomena have been studied using animal models, cell cultures, PET tracers, and histopathology. This methodological heterogeneity poses some problems in the interpretation of results. Although precise markers for MS have yet to be identified, as we previously mentioned, a particular secretory profile was discovered [37,38,39]. In particular, analyzed in both humans and mice studies, SASP consists of an enhanced secretion of chemical mediators of inflammation as well as those engaged in matrix degradation activities [3,56]. Recent studies documented that senescent microglia may release high levels of Tumor Necrosis Factor (TNF) and Interleukin-6 (IL-6) [57,58]. Moreover, SASP is associated with mitochondrial malfunction and subsequent abnormalities in energy metabolism, as well as increased Reactive Oxygen Species (ROS) generation and DNA damage [56,59,60,61,62,63]. In line with these findings, SASP may be considered a peculiar and complex microglial signature that is important to detect MS. Furthermore, Neumann and colleagues showed that MS is related to higher ferritin expression in the postmortem brain tissue of healthy controls [64]. The increase in ferritin goes along with patients’ age, increases oxidative stress, and is associated with morphological changes (de-ramification). Interestingly, they did not find other typical markers associated with oxidative stress, such as upregulation of 8-hydroxy-2′-deoxyguanosine (8-OHdG), heme-oxygenase-1 (HO-1), phosphorylated histone variant H2AX (y-H2AX), and lipofuscin deposits. Therefore, ferritin expression seems to be the dominant immunophenotypic change related to microglial aging and senescence [64]. Aging itself implies a “replicative senescence”, occurring during multiple replicative processes, due to telomere shortening. Telomere shortening in human microglia is a well described phenomenon, more evident in patients with AD [57,65,66]. As microglial cells are the scavengers of the protein debris related to neurodegeneration, the stimulus to replication induced by proteinopathies may promote MS. In this frame, Hu and colleagues compared postmortem brain tissue of patients with AD and healthy controls and observed that microglial proliferation in AD cases promotes telomere shortening and senescence, evidenced by increased beta-galactosidase activity (a senescence signature), with the appearance of “disease associated” microglia states [57]. In fact, β-amyloid (Aβ) accelerates MS in humans by upregulating genes coding for p21 and plasminogen activator inhibitor 1 (PAI-1) and beta-galactosidase activity [58].

Senescence and neurodegeneration lead to dystrophy; indeed, MS has certain morphological hallmarks that coincide with those of dystrophic microglia, characterized by abnormalities in their cytoplasmic structure, such as de-ramified, atrophic, fragmented, or unusually tortuous processes [67]. Shahidehpour and colleagues found that the number of dystrophic microglia was greater in cases with AD, Lewy bodies disease (LBD), or limbic-predominant age-related TDP-43 encephalopathy (LATE) compared with healthy controls [67]. Specifically, analyzing the brain tissue of patients with AD, Davies and colleagues observed that 49–64% of microglia displayed discontinuous and/or punctate Iba1 labeled processes, rather than continuous Iba1 distribution. On the contrary, only 16% of age-matched control microglia had discontinuous or punctate characteristics. Microglial cell body density did not differ between the two cohorts [68]. Iba1 is a cytoskeleton-associated protein expressed on microglial cell bodies and processes, so it is used as a microglial antigen to study both microglial cell number and morphology. These findings suggest microglia undergo a progressive cell process retraction with aging, but patients with neurodegeneration, and particularly AD, display further microglial morphological alterations, such as a reduced arborization in cortical gray matter compared to age-matched controls [68].

To date, it is still disputed whether “senescent” and “dystrophic” microglia are related to the same condition or are distinct. While biochemical and molecular markers of senescence exist, Streit et al. claim that there is no currently straightforward means to identify the dystrophic features of microglial cells, apart from morphological abnormalities, and that changes described in cell morphology cannot be easily correlated with modifications at the molecular level [35,39]. It has been suggested that over-reactive microglia can reach an exhausted phase, unable to escape their extensive faults in energy metabolism, resulting in “senescent” or “dystrophic” microglia [27,69]. Considering the above, we believe that microglial “senescence” is related to aging, is accelerated by neurodegeneration, and is one of the major mechanisms leading to the appearance of “dystrophic” morphology, which has no clear molecular signatures because it occurs in now exhausted cells (Figure 3).

### 4.2. Microglial Morphology, Nomenclatures, and States: Past and Future

Traditionally, microglial configurations and functional states have been characterized by the morphology, which remains of cardinal importance for histological analysis. Nonetheless, there is an increasingly broad spectrum of neurobiological parameters that may be used to define microglial functional states, such as those obtained through single cell transcriptome analysis (scRNA-seq) [8]. These studies have mainly been carried out on mouse models, which is a fact which raises the problem of the marked differences between humans and animal models, highlighting the need to have human brain tissue suitable for specific studies on humans.

Microglial states are determined by intrinsic variables (e.g., sex and genetic background), but also by the specific context in which they exist (e.g., age, anatomic location, and environmental factors). As previously suggested by Paolicelli and colleagues, these factors interact with microglia at numerous levels (epigenomic, transcriptomic, proteomic, metabolomics, and ultrastructural), eventually determining microglial configurations and activities [8]. Several microglial transcriptional profiles have been found in recent scRNA-seq studies, including proliferative-associated microglia (PAM) and axon tract-associated microglia (ATM) in early postnatal mice models; disease-associated microglia (DAM), microglial neurodegenerative phenotype (MGnD), and activated response microglia (ARM) in disease models of AD; and white matter-associated microglia (WAM), human AD microglia (HAM), and lipid droplet accumulating microglia (LDAM) in aging, both in mice and humans [8].

In the following paragraphs we try to clarify and summarize the most recent acquisitions in this regard, considering both the traditional morphological approach and the most recent acquisitions in terms of gene activation (scRNA-seq), distinguishing what concerns humans and animal models.

#### 4.2.1. Microglial Morphological Hallmarks

As regards humans, we found the following terminology to indicate the morphological patterns of the diverse activation states of microglia. In a nutshell, there are normal or “homeostatic” microglial cells and “active” or “reactive” cells including both “primed” and “ameboid” microglia. Regarding the terms “active” or “reactive”, we prefer the latter, since even homeostatic microglial cells are somehow always active. Homeostatic microglia correspond to a “ramified” morphology with a thin body and many ramified processes. In contrast, reactive cells are morphologically characterized by an enlarged body with fewer and stubbier processes. Also, in aged brains, there are peculiar microglial features such as “rod-shaped” cells showing elongated cell bodies with polarized processes. Furthermore, in aging as well as in AD, “dystrophic” features have been frequently observed corresponding to cells that are functionally depleted (Figure 3).

Ramified microglia are usually considered non-active or in a homeostatic state. However, the terms “non-active” or “homeostatic” do not adequately express the functional state of these cells, which are always engaged in some function. Therefore, it is much preferable to use the term “homeostatic microglia”. These cells are the “sentinels” of CNS but, at the same time, they actively participate in a variety of physiological processes during the normal brain functioning, including a phagocytic activity via their terminal or “en passant” branches, particularly during adult neurogenesis [70,71].

Primed microglia have a rounder and bigger cell body with a decreased arborization and complexity of ramified processes [68,72]. Microglial priming is a response to various microenvironment perturbations, representing the first line CNS defense (innate immunity) [72,73,74]. The increased metabolism of primed microglia may cause ROS generation and promote neurodegeneration [75,76]. Primed microglia are more sensitive to inflammatory microenvironments due to high levels exposure of IL-1, IL-12, and interferon. Interferon is thought to be involved in “classic” priming, which is neurotoxic in nature, whereas Toll-like receptors (TLRs) 2, 3, and 4 are thought to be involved in “alternative” priming, which is considered neuroprotective [77,78]. In the context of AD, microglial priming describes a phenomenon where continuous mild stimuli (e.g., protein debris and systemic inflammation due to aging) lead previously homeostatic microglia to enter a modified state, which is more sensitive to pathological triggers and may cause an excessive inflammatory reaction [18] (Figure 3).

Ameboid microglia have phagocytic properties and show few unramified processes. Tissue damage signals, including the release of potassium and ATP/ADP, activate the Purinergic Receptor P2Y12 (P2RY12; specific for microglia and not present on other macrophages), which potentiates the microglial tonically activation through the two-pore domain halothane-inhibited K + (THIK-1) channel, hyperpolarizing the cell and resulting in reduced microglial ramifications. These cells represent the most aggressive phenotype with phagocytic properties that may lead to neuronal death, neuronophagia, and phagocytosis of fibers, myelin sheaths, and damaged synapses, but also to the elimination of tissue debris. In this frame, a molecular relationship between microglial decreased branching and increased production of the inflammatory cytokine IL-1b has recently been discovered [79]. In turn, amoeboid microglia can change and lose their aggressiveness, presenting different functional and evolutionary patterns. For instance, ameboid microglia can exhibit impaired phagocytosis in some disease circumstances such as neurodegeneration and epilepsy [80], showing dystrophic features with fragmented and shorter processes, some of them still intact at the ultrastructural level [8] (Figure 3).

Rod-shaped microglia (elongated cell bodies with polarized processes) increase with age but are not associated with any particular neurodegenerative disease. The function of these cells has not been clarified but one hypothesis suggests that rod-shaped microglia could serve a protective role for neurons, possibly by encircling mildly damaged neurons to enhance their chances of survival [81]. Therefore, rod-shaped microglia may represent an attempt to protect senescent and dysfunctional neurons. On the other hand, there are dystrophic microglia showing progressive cell processes retraction and fragmentation, leading to a reduced arborization. These dysfunctional morphologies rise in parallel with aging, but are particularly prominent in patients with neurodegeneration and especially AD. Over the past two decades, the role of microglia has gained ever more importance in AD pathogenesis and microglial dystrophy has been shown to precede neurofibrillary degeneration in AD [27] (Figure 3). However, it is still not clear if microglial dystrophy is a cause or a result of AD; both hypotheses are probably true.

The investigation of microglial morphology is still regarded relevant and is frequently employed in animal models and human postmortem brain studies. We suggest that morphology is very useful for distinguishing homeostatic from reactive microglia but cannot clarify the particular functional state of reactive microglia; rather, the morphology represents a suggestion for further research on the link between microglial structure and function. Future studies are needed to show how various morphologies and antigenic patterns correlate with different transcriptional and proteomic profiles and what they imply for cell function.

#### 4.2.2. Microglial States in Mice Models of Neurodevelopment (PAM/ATM)

Early postnatal mouse models are experimentally used to study the role of microglia in neurodevelopment; nonetheless, in these models some microglial cells may present characteristics also typical of aging. Indeed, PAM and ATM phenotypes may share some features with the core DAM/WAM signature in mice and humans, typical of aging and neurodegeneration [82,83,84,85]. Probably, the processes of maturation and age-related “dematuration” share a number of common mechanisms.

Further investigations are needed to investigate similarities and differences between early PAM/ATM mice microglial phenotypes and DAM/WAM microglial states in aged mice and, possibly, humans. A representative scheme of all this spectrum of phenotypes is illustrated in Figure 4.

#### 4.2.3. Microglial States in Disease Models of AD (DAM/MGnD/ARM)

These microglial configurations include several reactive microglial phenotypes, showing characteristics associated with both M1 and M2 states, that are able to detect brain damage signals, including those related to AD, and stimulate phagocytosis and barrier building [49,86]. Typically, the spectrum of “reactive microglia” includes the DAM state, which is morphologically characterized by a de-ramified and amoeboid shape, with downregulation of various homeostatic genes such as *Transmembrane Protein 119* (*TMEM119*), *P2RY12/13*, *C-X3-C Motif Chemokine Receptor 1* (*CX3CR1*), *colony-stimulating factor-1 Receptor* (*CSF1R*), and *TGFβ*. On the contrary, *triggering receptor expressed on myeloid cells 2 (TREM2)*, *apolipoprotein E (APOE)*, *Major Histocompatibility Complex II (MHCII)*, *and Cluster of Differentiation 44 (CD44*) are all upregulated genes in DAM phenotypes [8,87,88,89]. Different DAM signature genes have been discovered in a variety of situations. A common set of markers are upregulated, including *TREM2*, *APOE*, *CD11c*, *CLEC7A*, and *LPL*, denoting a microglial state associated with several models of AD [8,90]. Furthermore, DAM exhibits a distinct pattern of localization involving areas more prone to neurodegeneration and AD pathology such as the hippocampus. These microglial cells exhibit enhanced phagocytic activity, which could be interpreted as an attempt to mitigate the disease, failing because it occurs at a relatively late stage of the disease. According to this conception, DAM microglia would not cause AD; on the contrary, they appear to be beneficial even though they are not sufficient to halt the progression of the disease [91]. Moreover, it has been proposed that DAMs activate in two stages, with TREM2 signaling mediating the transition from stage 1 to stage 2 [86,91,92]. The initial stage, which does not rely on TREM2, entails the activation of a specific group of genes, including the *TREM2-signaling adaptor TYRO protein tyrosine kinase binding protein (TYROBP)*, *APOE*, and *Beta-2-Microglobulin (B2M*). Simultaneously, it involves a decrease in microglial factors responsible for maintaining homeostasis, such as the receptors CX3CR1 and P2RY12/P2RY13. The subsequent phase of DAM activation, characterized by the stimulation of lipid metabolism and phagocytic pathways (e.g., *Lpl*, *Cst7*, and *CD9*), is TREM2-dependent. These two stages are represented by two ontogenetically distinct cell lineages, both expressing TREM2 and accumulating with age: resident microglia and invading monocyte-derived cells (termed “disease inflammatory macrophages”, DIMs) [8,93]. In the absence of the TREM2 receptor, this transition to fully activated DAM does not occur. Indeed, TREM2 is crucial to induce microglial phagocytic activity as in *Trem2* knockdown mice microglia show impaired phagocytosis and overproduction of inflammatory cytokines. On the contrary, TREM2 overexpression favors phagocytosis and has anti-inflammatory effects [94]. Anti-inflammatory activities related to TREM2 signaling are due to the inhibition of the TLR-4 through the c-Jun N-terminal kinase (JNK) and Nuclear Factor kappa B (NFκB, a group of transcription factors regulating numerous genes associated with immune and inflammatory reactions) signaling pathways [94,95,96]. These data support recent findings indicating that the absence of TREM2 in microglia during the late stage of AD exacerbates disease symptoms, whereas it does not seem to have the same effect during the early stage of AD [91]. *TREM2* mutations are infrequent but determine a neurodegenerative risk similar to that of the *APOE* ε4 allele (*APOE4*), the most important genetic risk factor for late-onset AD [97,98,99]. However, many concerns remain unanswered about the DAM signature’s functional significance and the transferability of these observations to humans. A representative scheme of this spectrum of phenotypes is illustrated in Figure 4.

#### 4.2.4. Human Microglial Reactive States (M1/M2 Paradigm and WAM/HAM/LDAM)

The plain M1/M2 paradigm has been conceived, using simple in vitro models, to comprise the various reactive microglial states that may have detrimental inflammatory effects (M1) or repair tasks (M2) (Figure 1). Both M1 and M2 microglia show a stocky and little branched morphology (reactive phenotype), but they have a very different spectrums of cytokine production. Indeed, cytokines with opposite activities may be produced by cells with similar morphological characteristics, confirming that it is not possible to deduce the kind of activation by only considering the morphology. Arguably, during the in vivo functioning of the human brain, M1 (producing TNF-α, IL-1α, IL-1β, IL-6, IL-12, IL-23, chemokine (C-X-C motif) ligand 9 (CXCL9), CXCL10, CXCL11, CXCL16, chemokine (C-C motif) ligand 5 (CCL5)) and M2 (producing IL-10, TGF-β, CCL1, CCL17, CCL18, CCL22, CXCL13, Vascular Endothelial Growth Factor (VEGF)) are at the two extremes of a series of subpopulations with intermediate secretory and functional characteristics, and the existence of an overlap between markers for distinct states runs counter to the binary “M1/M2” paradigm, particularly in vivo. Specifically, M1 is involved in antigen presentation and displays an upregulation of MHC II, CD86, Fcγ receptors, and inducible nitric oxide synthase (iNOS), while the M2 phenotype fails to capture all the diversities of neuroprotective microglial subpopulations. As a result, a shift towards M2a-c subclassification has been proposed (Figure 4). In this classification, the traditional alternative activation phenotype (neuroprotective), designated as M2a, is characterized by upregulation of Arg1 and scavenger receptors, along with inhibition of NF-κB isoforms. M2b identifies microglia involved in immunoregulation, while M2c is involved in tissue remodeling [100,101,102,103]. More recently, Paolicelli and colleagues suggested that the introduction of scRNA-seq technology offers the possibility of studying the different microglial states in more depth, overcoming the simplistic M1-M2 classification [8,46].

The core DAM signatures in mice models are partially shared by WAM in humans [8,82,104]. In particular, the WAM state relies on the activation of the TREM2 pathway and is influenced by the aging process. According to a recent human brain tissue investigation, TREM2 seems not to be expressed by homeostatic microglia but it appears as a hallmark of recruited monocytes [105]. On the other hand, in the aged human brain, the WAM profile forms independently of *APOE* [8,82,104]. This is in contrast to AD mouse models, where microglia exhibiting the WAM gene profile develop prematurely and in an APOE-dependent way, similar to the DAM state. In white matter regions, microglia often aggregate into nodules, where their primary role is to clear deteriorated myelin. Consequently, WAM may signify a potentially protective response essential for the clearance of degraded myelin that accumulates during the aging process and in cases of white matter-related diseases. Actually, the functional characteristics of human microglia are quite different from that of murine microglia, as demonstrated also by the relevant differences between murine DAM and human AD-associated microglia (*id est* HAM). Indeed, Srinivasan et al. analyzed 21 patients with AD and 21 healthy controls’ postmortem brain tissue, observing the particular expression profile of HAM [106]. They observed little in common with the DAM state. The HAM pattern expresses the human aging profile, including *APOE* overexpression and, particularly in patients with AD, upregulation of the *ABCA7*, *GPR141*, *PTK2B*, *SPI1*, *ZYX*, *MS4A6A*, *MS4A4A*, and *NME8* genes. On the contrary, these cells downregulated the expression of *MEF2C* and *CECR2*. Nevertheless, a common feature between DAM and HAM signatures was identified in the involvement of lipid/lysosomal biology-associated genes [106].

Marschallinger and colleagues named “LDAM” a new state of microglia in the aged brain in which they accumulate lipid droplets [107]. Using a mouse model, they highlight a distinct transcriptional signature, phagocytosis abnormalities, higher amounts of ROS, and higher levels of pro-inflammatory cytokines. Six genes with autosomal dominant neurodegeneration variants were identified: *SLC33A1*, *SNX17*, *VPS35*, *CLN3*, *NPC2*, and *GRN*. They showed that LDAM represents a dysfunctional and pro-inflammatory microglia state in the aging brain and that it may contribute to neurodegenerative disorders [107]. In these cells there was an upregulation of *PLIN3*, *ACLY*, *CAT*, and *KI* genes [8,107]. A representative scheme of all this spectrum of phenotypes is illustrated in Figure 4.

Perilipins proteins (Plin1-Plin5) decorate intracellular lipid droplets and they are important in lipid metabolism. Nevertheless, little is known about their expression in the human brain or their role in brain aging and neurodegeneration. A recent study investigated the expression levels of perilipins in different cerebral locations of participants of various ages, with and without signs and symptoms of neurodegeneration [108]. Specifically, Plin2, Plin3, and Plin5 are expressed at varying levels in the brain areas studied, but only Plin2, which is related to lipid droplets, appears to be controlled by age, neurodegeneration, and seems to be associated with IL-6 expression and neuroinflammation [108]. Further research in this area is required to better understand the relationship between brain aging, lipid deposits, microglia, and age-related disorders.

### 4.3. Homeostatic Microglia: Responses to External/Internal Perturbations and Aging

In agreement with what was reported by Gosselin, Galatro, and others, Böttcher and colleagues confirmed the primary transcriptomic markers of human microglia at the protein level, including the presence of P2Y12 and TMEM119, elevated expression of CD64, CX3CR1, TGF-β, TREM2, CD115, CCR5, CD32, CD172a, and CD91, and minimal or no expression of CD44, CCR2, CD45, CD206, CD163, and CD274 (PD-L1) [109,110,111]. They found that the immunophenotypes of “postmortem” human microglia of a heterogeneous group of individuals (e.g., epilepsy patients, young and old healthy controls) were similar to “fresh” human microglia, although there were variations in signal strengths for certain markers. Moreover, they found that human microglia display various phenotypic characteristics based on the specific brain area they occupy. These region-specific differences might be due to the fact that microglia perform different functions depending on the region in which they are located [111]. Microglial reactivity in response to injury or noxious insults is localized, rather than engulfing the entire brain8 because the reaction occurs at the site of the lesion and it also depends upon microglial regional characteristics [3]. Masuda and colleagues found similar results by identifying microglia subtypes with different gene expression profiles depending on the brain region they are located in [7]. Moreover, Böttcher and colleagues found microglia at different phases of the cell cycle and an increased proliferation of microglia in the thalamus and in the subventricular zone, which could reflect a region-specific function of microglia. They also observed that the primary immunophenotype differentiating resident microglia from circulating and invading macrophages persists across the different postmortem human brain areas examined [111]. Grabert K. and colleagues pointed out regional differences in the microglial metabolism [112]. Particularly in mice, they found regional differences in the transcription of genes involved in microglial metabolism: microglia in the hippocampus and cerebellum have a higher expression of genes involved in mitochondrial biogenesis, glycolysis, Krebs cycle, electron transport chain, and ATP creation, pointing to a greater energy need in these regions [112].

Homeostatic microglia consistently monitor the CNS microenvironment in physiological conditions; they are involved in the activation of the first line non-specific reaction to foreign antigens (pathogen-associated molecular patterns—PAMPs) and to the subsequent antigen presentation to lymphocytes inducing the specific immunological response. They are the most important antigen-presenting cells (APCs) in the CNS. Moreover, homeostatic microglia react to internal perturbations (damage-associated molecular patterns—DAMPs). PAMPs and DAMPs bind to pattern recognition receptors, which include Toll-like receptors (TLRs) [113,114]. PAMPs are derived from microorganisms and therefore drive inflammation in response to infections [115]. A paradigmatic example of PAMP is Gram-negative bacterial lipopolysaccharide (LPS). Instead, DAMPs arise from host cells, including tumor cells, dead or dying cells, protein debris, or products released by cells in response to damage signals, such as hypoxia, metabolic suffering, or other severe dysfunctions. DAMPs arise from a non-specific response to non-infectious tissue damage [115,116]. Cell types that express pattern recognition receptors include innate immune cells such as monocytes, macrophages, and precisely the microglia [113]. Pattern recognition receptor–ligand binding initiates conformational changes inducing a downstream signaling cascade that results in the transcriptional modifications and post-translational and morphological changes. In humans, the reaction to PAMPs leads to clinical manifestations including “sickness behavior”, lethargy in cases of severe inflammation, up to delirium in cases with neurodegenerative burden. This is the result of an increased production of proinflammatory cytokines by microglia resident in the brain parenchyma, comprising IL-1 (both IL-1α and IL-1β), IL-6, and TNF-α. Peripheral inflammatory signaling influences the brain through two principal routes: a neural pathway through afferent neurons connected to the body area undergoing infection or inflammation, and a humoral route through the blood circle, mainly mediated by the choroid plexus and venular system [117]. Other authors have demonstrated that acute systemic inflammation has an effect on microglia and contributes to a microglial state associated with robust IL-1β production [118,119,120]. In humans, paradigmatic evidence of the link between systemic infections and inflammation, and microglial activation has emerged from neuropathology studies in Coronavirus disease 2019 (COVID-19). COVID-19 brains appear to have a microglial overreaction, especially in brainstem and hippocampus, leading to neurological manifestations [121]. Yang and colleagues analyzed the transcriptome of 65,309 nuclei of brain cells belonging to the frontal cortex and the choroid plexus of 30 patients who died with COVID-19. They detected an increased expression of the *IFITM3* gene in glial cells and in the choroid plexus, as well as an increased expression of molecules involved in the inflammatory response, such as genes coding for interferons and complement factors. This study is interesting as it highlights a link between systemic inflammation and brain inflammation, with the inflammation of the choroid plexus being the communication point. Activated by systemic inflammation, the choroid plexus produces chemokines that trigger an inflammatory response at the level of microglia, astrocytes, oligodendrocytes, and some cortical excitatory neurons [122]. While the word “neuroinflammation” is extensively used as a synonym for microglial reaction, its definition differs substantially among authors. The term “neuroinflammation” should not be used interchangeably with “microglial reaction”. Indeed, microglial reaction is a part of neuroinflammation which involves intricate local responses and interactions with all the cells involved in the immune response, including microglia, astrocytes, oligodendrocytes, and endothelial cells. The reaction of brain parenchyma to an injury implies various transcriptional states of these cells, whose implications are not fully understood. Not all these states may be inflammatory, some might be homeostatic or non-inflammatory reactive states. Moreover, neuroinflammation is not always harmful. Its effects, whether adaptive or maladaptive, depend on the context.

It is well known that microglial cells respond not only to local cues within the brain but also to continuous inputs from the periphery, such as signals from the gastrointestinal tract [123]. Enry et al. found in mice that a “less diverse” microbiota leads to malfunctioning microglia and that short-chain fatty acids (SCFAs), which are products of bacterial fermentation from the microbiota, play a role in maintaining the balance of microglia. SCFAs might move from the intestinal lining into the bloodstream, potentially influencing immune system regulation and the functioning of the central nervous system [124]. Although there is much evidence regarding external and physiological factors influencing microglia in mice, little has yet been explored about it in human microglia. Also, sex-related differences in microglial response are explored in mice but they are not clearly defined in humans. In mice models, sex differences caused by sex chromosomes and/or gonadal hormones may have an impact on microglial states in certain settings (e.g., in response to PAMPS). The sex-specific gene expression profiles of microglia appear to be intrinsically defined as they are retained when microglia are transplanted into the brains of mice of the opposite sex [125].

Over time, through the individual’s personal history, the effects of external and internal perturbations profoundly influence the microglial structure, which in turn may be protective against neurodegeneration or favor neurodegenerative diseases. Indeed, aging determines a cumulative increase in the perturbations faced by microglia. In mice, it has been found that aging microglia show reduced transcriptomic diversity, even across different brain regions [85]. As mice models age or suffer from neurodegenerative disorders, microglia exhibit a decreased efficiency in quickly addressing brain stressors, such as eliminating harmful amyloids or managing infected, harmed, or declining neurons. This can contribute to CNS dysfunctions and to the further progression of diseases [85]. Considering the effects of aging on microglia, in recent years there has been a growing interest in developing strategies for the eradication of senescent microglia as a means of remediation for aging and neurodegeneration. Such experimental models added a deeper knowledge of key factors involved in microglia-related pathways [3,126]. For example, CSF1R was targeted for treating phospho-TAU propagation in AD. CSF1R has been shown to be crucial for microglial development and survival [3,126]. Two cytokines, IL-34 and CSF1, support this pathway by binding to CSF1R. IL-34 is produced by neurons, while astrocytes and oligodendrocytes mainly secrete CSF1 [127]. CSF1 mainly aids in the development of microglia in areas like the corpus callosum, pons, and cerebellum. On the other hand, IL-34 supports the sustenance of microglia in the forebrain regions, but it does not do so in the cerebellum or brainstem [127]. Targeting CSF1R has proven effective in eliminating up to 99% of CNS microglia in mice. When the inhibition of CSF1R is lifted, microglia can repopulate either from proliferation of remaining microglia or potentially from a progenitor, although the existence of such a progenitor is debated. Though CSF1R knockout mice lack microglia and die young, adult mice undergoing depletion and repopulation show no cognitive or motor deficits. The repopulated microglia closely resemble the original microglia. Interestingly, depleting and repopulating microglia in older mice improves cognition and boosts synaptic spine density and neurogenesis [3,126]. In another model, microglia depletion was obtained with mice expressing the herpes simplex virus encoding the “suicide-gene” thymidine kinase (HSVTK) driven by the CD11b-promoter. Giving ganciclovir led to up to a 95% depletion of microglia, but it became toxic after prolonged use, limiting its use to 4 weeks. When ganciclovir was stopped, the microglial pool was replaced by peripheral myeloid cells [3,126]. A further step was the creation of CX3CR1^CreER^ mice, which, when paired with Rosa26^DTR^ mice and treated with tamoxifen and diphtheria toxin, effectively depleted microglia without impacting bone marrow-derived CX3CR1+ cells. However, as other CNS-associated macrophages also have CX3CR1, they might also be depleted in these mice [3,126]. The effectiveness of such strategies could be increased by a more precise identification and targeting of senescent microglia, which would necessitate the discovery of a unique, specific marker [3]. Transcriptional analysis identified unique microglia genes like *Spalt-like Transcription Factor 1 (Sall1)*, leading to the development of Sall1^CreER^ mice that target microglia without affecting peripheral and CNS-associated macrophages, but these mice have not yet been used for microglia depletion [126]. Another potential alternative treatment to slow down or stop the senescence of microglia could be the genetic and pharmaceutical rescue of Sirtuin-1 (SIRT1) pathway. Aβ accelerates microglial senescence. SIRT1 is an NAD-dependent deacetylase that regulates cell senescence, metabolism, inflammation, and mitochondrial function. Under normal conditions, SIRT1 expression and activity are tightly regulated [58,128,129,130,131]. However, its expression declines during aging, metabolic disturbances, or neurodegenerative diseases, potentially amplifying oxidative stress [132,133,134]. A significant decrease in SIRT1 levels is associated with the buildup of Aβ and TAU proteins in patients with AD. Some evidence suggests that SIRT1 can mitigate Aβ deposition and toxicity, improving AD pathology [108,135,136,137,138,139]. Moreover, SIRT1 is intricately linked with nuclear E2-related factor 2 (NRF2), a transcription factor that modulates inflammatory responses [140,141,142]. Thus, targeting the SIRT1/NRF2 pathway could be a potential strategy for combating neurodegenerative diseases related to aging. It is noteworthy that in AD mice models and patients, SIRT1 protein levels are considerably reduced, correlating with increased Aβ and TAU proteins compared to normal aging individuals. Interestingly, aspirin, a common anti-inflammatory drug, was observed to boost SIRT1 levels and counteract microglial senescence-Aβ induced in vitro [58] Although aspirin reduced amyloid plaque in an AD mouse model, it was not demonstrated to reduce Alzheimer’s risk in clinical trials [58]. While targeting and reprogramming aged microglia are very interesting from the point of view of scientific speculation, their transferability into human treatment protocols remains highly uncertain. Future treatments would ideally target only the harmful microglia without requiring genetic manipulation.

### 4.4. Interactions of Human Microglia with Aβ and pTAU

AD pathology rises by the association of Aβ deposition (senile plaques) and phospho-TAU (pTAU) aggregates (neurofibrillary tangles—NFT, and threads). The combination of these two proteinopathies in the neuritic plaques (NPs) constitutes the pathological hallmark of AD. Also, hypoxic brain suffering due to small vessel disease is frequently associated with neurodegeneration and could impact both degenerative processes and microglial functions [143]. Some years ago, Di Patre and colleagues revealed that NFT counts were substantially linked with reactive microglial cells counts in human AD, particularly within the subiculum. They observed that the burden of reactive microglia correlated significantly with the burden of NFT (*p* < 0.005), but not with senile plaques that do not contain pTAU [144]. Nonetheless, more recently, it has been found that the proportion of reactive microglia in postmortem cortical tissue is strongly associated not only with pTAU neuropathology but also with Aβ aggregates [84]. Furthermore, some data indicate that microglial overactivation may contribute indirectly to cognitive decline, favoring the accumulation of pTAU [84]. Indeed, Pascoal et al. showed that reactive microglia boost the pTAU spread across Braak stages and Aβ potentiates the effect of microglia on pTAU spreading. Co-occurrence of Aβ, pTAU, and microglia abnormalities was in fact the strongest significant predictor of cognitive impairment (*p* < 0.0001) [145]. In line with these findings, previous studies suggested that microglial activation may be required for pTAU spreading via exosome release, and changes in microglial phenotypes may thus contribute to pTAU pathogenesis, leading to altered TAU proteostasis [146,147]. TAU pathology is pivotal in AD pathogenesis as there is clear evidence supporting the synaptic toxicity of tau oligomers. Indeed, Singh and colleagues found scarce presence of TAU oligomers in the synapses of subjects with AD neuropathology but resilient to dementia [148]. An abnormal buildup of soluble hyperphosphorylated TAU within synapses was observed in AD. These soluble species seem to be more detrimental than NFTs for synaptic function, leading to the clinical symptoms associated with AD [149]. The removal of damaged synapses constitutes a fundamental activity to maintain brain homeostasis, in which microglia are engaged. Considering that astrocytes are prevalent in microglial cells, and that the majority of synapses are closely associated with astrocytes [150], they are strongly involved in synaptic remodeling in cooperation with microglia. Then, it is possible that astrocytes have a bigger impact than microglia on synaptic loss in the first stages of AD. Both microglia and astrocytes can phagocytize damaged synapses in the human brain, especially in presence of TAU soluble oligomers, even without a clear presence of NFT. An excessive removal of synapses leads to brain dysfunction and brains that show resilience to dementia present a decreased engulfment of synapses containing TAU oligomers, potentially explaining the maintained cognitive abilities in these individuals [151].

As for the relationship between Aβ and microglia, several recent studies on rodents emphasized this interrelation [3,8,152,153]. In mice microglia cultures, Dhawan et al. [55] observed that oligomers activated microglia, leading to an inflammatory response, which in vivo could be protective or harmful, or both in different time frames of disease evolution. This process occurred via a pathway associated with tyrosine kinase, resulting in the secretion of TNF-α. These findings were validated both in primary murine microglia cultures and in live animals. Specifically, in AD models and in human brains, an increase in reactive microglia was observed, associated with an increase in phospho-tyrosine, p-Lyn, and p-Src levels. Moreover, the intracerebroventricular infusion of oligomeric Aβ increased microgliosis via a tyrosine kinase-dependent mechanism that was attenuated by the administration of dasatinib (a tyrosine kinase inhibitor). The drug can cross the blood–brain barrier and was proposed as a possible candidate for AD treatment [152]. But are these results from experiments on rodents related to what happens in humans? Recently, Wang and co. created a mouse model to study the signaling of the defective *TREM2R47H* human variant [154]. Normally, microglial cells close to Aβ plaques display a DAM transcriptional signature deriving from the TYROBP receptor complex, which transmits intracellular signals by means of the spleen tyrosine kinase (SYK). The human *TREM2R47H* variant, carrying a high AD risk, fails to activate microglia through SYK. The authors observed that microglia lacking SYK activation could not encapsulate Aβ plaques accelerating AD pathology and cognitive deficits and demonstrated that immunotherapies boosting SYK through C-Type Lectin Domain Containing 7A (CLEC7A) improved microglia activation [154]. Moreover, in aging and in the presence of a pro-inflammatory frame, as in areas affected by AD pathology, microglia are less efficient in removing Aβ [101]. Regarding the relationships between mouse models and human pathology, we report here other interesting studies. A recent study analyzed the role of the Dickkopf WNT Signaling Pathway Inhibitor 2 (DKK2) in mice and in postmortem brain tissues, finding that *DKK2*+ microglia clustered near Aβ plaques in mouse models but not in human microglia [155]. *DKK2* is a modulatory gene of the WNT signaling pathway, upregulated downstream of TREM2, which is involved in the change from a homeostatic state to a DAM/ARM state through the activation of several genes involved in proliferation and survival. As upregulation of *DKK2* was not confirmed in human brains, the authors highlighted the limitations of relying solely on animal models to understand human neurodegenerative diseases [155]. On the contrary, APOE expression was found to be higher in microglia surrounding β-amyloid plaques, corresponding the DAM phenotype in both human and animal models. The DAM phenotype, which is observed in various neurodegenerative models and appears after the microglia engulf dying neurons, is dependent on the TREM2-APOE pathway [87]. Apoptotic neurons activate the TREM2-APOE pathway and suppress TGFβ signaling, which regulates the homeostatic signatures. In addition to TGFβ, APOE signaling suppresses SMAD3, MEF2a, and PU.1 signaling [87]. PU.1 is an essential lineage-dependent transcription factor (LDTF) needed for all macrophage subsets, and it works by selecting both common and cell-specific enhancers through interactions with other transcription factors. The areas where PU.1 binds become specific sites for different signal-dependent transcription factors (SDTFs) to act [156]. Furthermore, the TREM2-APOE pathway regulates core microglial genetic signatures, through a molecule called miR-155; the activation of this pathway might lead to DAM microglia losing their protective properties. For this reason, some authors suggest that manipulating the TREM2-APOE pathway could be a potential therapeutic approach to restore the homeostatic state of microglia and possibly treat neurodegenerative diseases [87].

The relationship between APOE genotype, TREM2, and microglial behavior is crucial for AD pathogenesis. Indeed, elevated APOE4 levels in cerebrospinal fluid were associated with cognitive decline [157]. To study the relationships between APOE, microglia, and AD pathogenesis, cellular models (iPSC) and animal models were used; only more recently have studies been carried out directly on human tissue. For example, studies on APOE4 through iPSC astrocytes and microglia-like cells were particularly focused on the molecular interrelations between Aβ accumulation and dysregulated cholesterol metabolism including increased cholesterol synthesis, lysosomal cholesterol sequestration, and decreased cholesterol efflux that are related to APOE4 genotypes [158]. Interestingly, when the conditioned medium from astrocytes carrying APOE4 was added to neurons, a higher secretion of Aβ42 and increased APP expression was observed [159]. The impact of APOE4 was also described in iPSC-derived brain cell types at the transcriptomic level: microglia-like cells had more dysregulated genes than those found in iPSC neurons and iPSC astrocytes and mostly were implicated in immune system processes, resembling an AD-associated pattern [160]. Moreover, two recent works in mice models demonstrated that APOE4 microglia upregulated homeostatic drivers, including TGFβ and PU.1, reducing their capability to digest amyloid plaques and other neurodegenerative debris. This observation was confirmed in human tissue: brains carrying APOE4 showed less reactive microglia around plaques [161,162]. A confirmation of the role of APOE4 and lipid dysregulation in AD pathology has been recently provided using a mouse model carrying human APOE4. This study demonstrated the efficacy of a liver-X receptor (LXR) agonist in boosting the efflux of lipids from microglia that reduced inflammation, p-TAU generation, and neurodegenerative phenomena. Unfortunately, LXR agonists are not a viable therapy, due to severe side effects, but a potential therapeutic target has been identified [163]. Another recent study on mice and human-derived iPSC showed that the APOE-R136S mutation antagonizes the deleterious effect of APOE4 on microglial function, providing protection against the development of the AD pathology and suggesting a possible therapeutic target [164].

Other interesting studies, carried out in humans, concern the role of hypoxia, often present in association with AD pathology due to coexisting micro-vascular damage. In this frame, March-Diaz and colleagues found that systemic persistent hypoxia activates the hypoxia-inducible factor 1α (HIF-1α) pathway in human reactive microglia of both healthy controls and patients with AD, favoring the evolution of AD by reducing the capacity of microglia to proliferate and contain Aβ deposits. In particular, they observed a significant microglial depopulation in severe AD (Braak stage: V–VI), particularly in the molecular layer of the dentate gyrus (hypoxia prone region) compared with the perirhinal cortex (control region) of the same individuals [165].

On the other side, Walker and co. investigated the behavior of the homeostatic microglia marker P2RY12 in relation to the presence of amyloid and NPs [166]. They observed that P2RY12 was not expressed by microglia near NPs in AD brains. Comparing the expression of P2RY12 in subjects without dementia (low NPs burden) with patients with AD (high NPs burden), Walker and co. pointed out that areas of inflammation are located around mature β-amyloid plaques and characterized by a paucity of P2RY12-positive microglia, while many diffuse plaques displayed co-localization with P2RY12-positive microglia [166]. Indeed, diffuse amyloid plaques may be part of a physiological aging pattern and do not induce an inflammatory reaction, which, instead, may be triggered by dense or cored plaques. Although P2RY12 seems to be a marker of homeostatic microglia, it is not completely clear whether microglia expressing P2RY12 are protective or pro-inflammatory. P2RY12-mediated responses may be involved in the early stages of inflammation, helping microglia to migrate to damaged or dying cells. Moreover, contrary to the belief that AD neuroinflammation is widespread, this study indicates that inflammation might be highly localized around specific areas, like mature amyloid plaques and NPs. The exact role of P2RY12 in AD remains complex and requires further research [166].

### 4.5. Resident Microglia: Differences with Circulating Monocytes, Transcription Factors, and Interleukins in Healthy Controls and Alzheimer’s Disease

During embryogenesis, resident microglia are irregularly distributed, and separate populations emerge during the early stages of development, with different transcriptomic phenotypes in both mice and humans [22,167]. In mice, it has been demonstrated that the transcriptomic phenotype of microglia is heavily dependent upon the local microenvironment [168], and more recently these findings have been replicated in human microglia [111]. Notably, Böttcher and colleagues discovered differences in the expression of the G-protein-coupled purinergic receptor P2Y12, TMEM119, and Interferon regulatory factor 8 (IRF8) as typical traits of human resident microglia [111].

As for mice, resident microglia are currently thought to be derived from a pool of macrophages generated during primitive hematopoiesis in the yolk sac and which begin penetrating the neuroepithelium at E8.5 (days old) [169,170]. On the other hand, in humans, resident microglial precursors infiltrate the brain primordium between 4.5 and 5.5 weeks of gestation [171,172,173,174,175]. In humans, there are two populations of resident microglia in CNS, which are anatomically separated: microglia proper and CNS-associated macrophages [7,176]. Microglia proper have their home within the brain tissue. On the contrary, CNS-associated macrophages are found in the perivascular space, choroid plexus, and leptomeninges; their identity is established early in development from yolk-sac-derived progenitors, and they can adopt a multitude of states depending on the different spatiotemporal context [8]. However, there are minimal differences between circulating monocytes that will become macrophages and human resident microglia, in terms of phenotype biomarkers and morphology, even though there are some markers specific for microglia proper such as P2Y12 and TMEM119. In fact, in the reactive phenotype state, the two populations are difficult to differentiate from a histological and immunohistochemical point of view. Furthermore, proper microglia and perivascular monocytes–macrophages interact and collaborate to maintain brain homeostasis. For example, macrophages contribute to the removal of perivascular amyloid deposits and their inefficiency could lead to amyloid angiopathy [177].

Now considering transcription factors and interleukins, the master transcription factors PU.1, IRF8, and SALL1 have been revealed to be crucial, respectively, for microglial formation, cellular lineage commitment, and maintaining microglial identity [9,173,178,179]. In particular, bone marrow-derived monocytes can differentiate into microglia-like cells, but they lack SALL1 expression and show some differences in cell shape [180]. As for resident microglia in human AD pathology, already a few decades ago Sheng et al. found that an increase in microglial activation and IL-1 expression with age may lead to an increased risk of AD. Specifically, they found that tissue IL-1α mRNA levels were higher in individuals over 60 than in those less than 60 years old (*p* < 0.05) and that activated IL-1α+ microglia may be classified into three morphological subtypes representing successive stages of activation: primed, enlarged, and phagocytic microglia [25]. A more recent study investigated the cytokine profile of the entorhinal cortex and the superior temporal sulcus belonging to cases resilient to dementia, despite showing intermediate (Braak stage: III–IV) or high AD pathology (Braak stage: V–VI). The profile demonstrated an upregulation of cytokines that have been associated with the clearing phase of inflammation, including IL-1β, IL-6, IL-13, IL-4, IL-10, and Interferon gamma-induced protein 10 (IP-10). All these cytokines were upregulated in cases resilient to AD in comparison with typical AD cases and controls. Moreover, resilient cases displayed a lower expression of chemokines associated with microglial recruitment, in particular Monocyte chemoattractant protein-1 (MCP-1) and Macrophage inflammatory protein-1 (MIP-1), in association with an enhancement of neurotrophic factors. This peculiar cytokine pattern corresponds to a different inflammatory activity that characterizes cases resilient to pathology [181].

Additionally, from our results it emerges that further studies have investigated the role of chemokines/cytokines and interleukins in human microglial cells (Table 1a,b shown at the end of the text). Specifically, some studies have highlighted the strong correlation between some chemokines/cytokines production (e.g., CXCL1, CCL3) and neuroinflammation, having a probable role in the differentiation from homeostatic human microglia to primed or LDAM phenotypes [107,145]. Other studies found that the production of some interleukins (e.g., IL-4, IL-6, IFN- γ) is associated with multiple microglia phenotypes, such as DAM/HAM, M1, and senescent/SASP [19,166,182,183]. Other recent evidence highlights the importance of IL-1α and TNF-α to stimulate microglial proliferation and promote primed phenotypes, while IL-1β is more related to a senescent/SASP microglial phenotype [57,58,184]. Moreover, a singular association between IL-33 expression and senescent microglial phenotypes has been recently observed [185], as well as a decrease in the inositol polyphosphate-5-phosphatase D (INPP5D) in microglia from human brain tissue favors the firing of the “inflammasome” in AD [186].

### 4.6. Human Microglia, Ferritin, Lysosomal Storage, and Mitochondrial Dysfunction

Microglia play an important role in iron homeostasis in the brain, absorbing and storing iron molecules by the protein ferritin [187]. Excess iron storage is considered a fundamental hallmark of elderly or senescent microglia [62,188]. Metal ion dyshomeostasis, in particular iron, has been identified as a contributing factor to microglial dysfunction [189,190,191,192,193,194,195]. Moreover, recent evidence demonstrates that iron supplementation [196,197] has been shown to cause cellular senescence, mimicking iron accumulation in the aging brain [39,190], with not all cell types similarly affected [198,199]. In particular, iron supplementation in vitro could be a useful tool to study human microglial senescence [3]. Heme—oxygenase 1 (HO-1), an enzyme which converts heme to biliverdin and iron, is induced in AD, pointing to an abnormal heme metabolism in neurodegeneration, and could be used as another possible important signal of degenerative phenomena [189,200]. Iron is a crucial cofactor for mitochondrial function and many enzymes containing heme are present in mitochondria. Proper iron metabolism is critical for energy production, possibly revealing a link between iron dyshomeostasis and altered energy metabolism in senescent microglia [3,201,202].

Mitochondria use a significant portion of the cell metabolically active iron, whose storage mainly occurs in the cytoplasm. The mechanisms mitochondria employ to maintain iron balance and prevent iron-related toxicity are not entirely clear. A specific gene on the chromosome 5q23.1 produces mitochondrial ferritin, an iron-storage molecule. When overexpressed, mitochondrial ferritin efficiently incorporates iron, potentially even more effectively than the common ferritin H found in the cytoplasm [187].

McCarthy et al. found that immortalized microglia can intake iron, both in transferrin-bound (TBI) and non-transferrin–bound (NTBI) forms. Microglia adapt their iron intake mechanism depending on environmental cues, suggesting that they may play a key role in managing brain iron levels and influencing neurodegenerative conditions: LPS (a component of bacterial walls) boosts the uptake of NTBI iron by microglia; this might be a strategy to deprive pathogens of iron. Moreover, inflammation prompts microglia to store the iron within the cell to prevent potential harm; in fact, under pro-inflammatory conditions, microglia mainly rely on the NTBI uptake mechanism. Conversely, under anti-inflammatory conditions induced by IL-4, microglia favor the TBI uptake route, reflecting a shift in their metabolic needs [202]. Ferritin expression has been reported to be enhanced in different neurodegenerative diseases [203], and it has been investigated as a potential CSF biomarker for predicting outcomes in AD [204]. Increased brain iron may lead to increased oxidative stress [191,194], and microglia, which seizes excess iron, has a neuroprotective role but may become vulnerable to oxidative damage. The overlap between ferritin-positive microglia and microglia with a dystrophic morphology in human patient samples suggests that iron buildup has a cytotoxic effect on microglia [36,192,194,205]. The increase in brain iron causes an upregulation of microglial ferritin in an attempt to prevent iron-mediated oxidative damage to neurons. The paradox is that also microglia is susceptible to iron-mediated oxidative damage, contributing to microglial dystrophy and dysfunction. Iron also encourages amyloid aggregation and oxidation, which in turn accelerate microglial dystrophy [194,206]. Moreover, microglial interaction with proteins involved in iron homeostasis is not necessarily negative; in fact, transferrin has been shown to increase microglial phagocytosis in the presence of a demyelinating lesion and at the same time could take part in remyelination processes, participating indirectly in oligodendrocyte maturation [207].

In mouse models, a novel type of microglia has been described [208]. These cells displayed numerous ultrastructural features of oxidative stress, giving them a “dark” appearance similar to that of mitochondria in electron microscope analyses. Indeed, they presented electron dense nucleoplasm and cytoplasm, dilatation of Golgi apparatus, and endoplasmic reticulum, along with cytoplasmic shrinkage. Dark microglia were found only sporadically under normal conditions in the hippocampus, cerebral cortex, amygdala, and hypothalamus, but their numbers increased significantly in situations of chronic stress, natural aging, deficiency in fractalkine signaling (CX3CR1 knockout mice), and AD pathology (APP- PS1 mice). Dark microglia were involved in a pathological reshaping of neural circuits engulfing dendritic spines, axon terminals, and even entire synapses. This type of microglia might be a subgroup of highly active microglia which become stressed when dealing with challenging situations, possibly manifesting oxidative stress. On the other hand, even if they did not express 4C12, which is a marker of inflammatory monocytes, they could develop from bone marrow-derived cells, recruited into the brain under pathological conditions. Actually, these cells can be recognized only by electron microscopy and the current markers do not allow for them to be distinguished from other myeloid cells [208].

As for human studies, comparing postmortem brain tissue of five patients with AD and eight healthy controls, Zeineh and colleagues discovered several tiny magnetic resonance imaging (MRI) hypointensities in AD cases, mostly in the subiculum, that were best described by a combination of iron deposits and active microglia assessed by histology. The subiculum is a crucial site where neurons from the entorhinal pathways intersect on their way to the hippocampus. Indeed the observed microglial presence may potentially indicate entorhinal neurodegeneration, a significant early-stage indicator of AD [209].

Burns et al. utilized cellular autofluorescence (AF) to investigate the diversity of microglia in physiological conditions in both rodent and non-human primate species. AF is the property of cells and tissues to give off light of a specific color once they have taken in light of a different color. Electron microscopy has shown that microglia exhibit AF due to the presence of lysosomal storage bodies inside the cell, which accumulate materials like fat molecules, cholesterol crystals, and other debris. Auto-fluorescent microglia have greater amounts of proteins associated with waste storage and digestion than those that are not auto-fluorescent. As the brain gets older, the buildup of lysosomal storage material in AF microglia intensifies and compromises the efficiency of the microglial activity, possibly due to mitochondrial dysfunction [188]. Mitochondrial DNA damage has been found to be increased in aged microglia due to the cumulative addition of stressing events such as those mentioned above [210,211].

### 4.7. Human Microglia Characterization: IHC and Related Phenotypes/Transcriptomic Profiles

Microglia are currently identified by the expression of particular genes that are substantially enriched in microglia, which represent their transcriptional identity; subsequently, they translate into a specific protein/receptor and phenotypic pattern depending upon the context. There are many antigens used as microglial markers in immunohistochemistry (IHC), especially for neuropathological investigations. The most frequently used IHC markers are CD68, HLA-DR, and Iba1 [13,14,15]. CD68 is predominantly expressed by intracellular lysosomes, but also on the cell membrane, and acts as scavenger receptor for oxidized low-density lipoproteins; HLA-DR is part of MCHII and is mainly expressed on the outer cell membrane; Iba1 is a cytoskeleton protein involved in cell motility and phagocytosis. Their cellular localization makes HLA-DR and Iba1 more appropriate than CD68 for morphological evaluations [212].

To some extent, HLA-DR, CD68, and Iba1 markers are present in all microglia phenotypes; nonetheless, these markers can be predominantly associated with different microglial states, including homeostatic (Iba1), primed/ameboid (both CD68 and HLA-DR), and senescent (both Iba-1 and HLA-DR) [64,127,175,176,177]. Interestingly, Iba1 staining was weakest in foamy macrophages in comparison to ramified and amoeboid microglia.

In 2017, Hendrickx and co. studied the gray matter of AD cases, observing that the frequency of Iba1-stained cells was modestly enhanced, but an increased expression of HLA-DR was found, particularly in advanced Braak stages. HLA-DR levels were higher than CD68 and Iba1 in microglia associated with high dense/cored amyloid plaque burden [212]. The limit of these markers is that they are not able to identify a specific microglial phenotype between the pro-inflammatory/anti-inflammatory spectrum and they are not related to a specific transcriptomic profile. Moreover, these markers are not exclusively expressed by resident microglia proper but are also present in macrophages. Considering human resident microglia, the two most specific known markers are P2RY12 and TMEM119, while CX3CR1 and TREM2 may also be expressed by macrophage-derived microglia (Table 2 shown at the end of the text). P2RY12 and TMEM119 are among the most promising markers of resident microglia proper. Also, P2RY12 is being investigated as a possible PET tracer [125].

Other markers have been studied, in particular the spectrum of CD11a, CD11b, CD11c, IL-2R, and CD163. Some of these markers are associated with peculiar morphologies or well-defined transcriptomic profiles. For example, Walker and colleagues found that CD11a, CD11b, CD11c, and IL-2 R could be potential human microglial markers of vacuolization after interaction with β-amyloid in brain tissue from healthy elderly controls [213]. Additionally, the CD11b marker was also reported in postmortem brain tissue of healthy controls and AD cases, specifically as a biomarker of HAM/DAM phenotypes in association with a higher expression of TMEM119 and P2RY12 and upregulation of *APOE*, *ABCA7*, *GPR141*, *PTK2B*, *SPI1*, *ZYX*, *MS4A6A*, *MS4A4A*, and *NME8* genes [102,125,148]. Moreover, CD45 [125,178], CD163 [209], and HLA-DP, -DQ, -DR [214] surface antigens were also used to classify primed/reactive human microglia. TMEM119 expression was also used as a marker of senescent/dystrophic human microglia associated with a downregulation of *GPR34*, *RASAL3*, *SASH3*, *ADORA3*, *CPED1*, *CIITA*, *IGSF6*, *LY86*, *LAPTM5*, and *P2RY13* genes, during aging; instead, *ACY3*, *ALOX5AP*, and *TLR7* genes are upregulated in AD [215].

Hence, despite the multiple attempts to match various markers to specific human microglial phenotypes or transcriptomic profiles, our systematic review still reveals that possible correlations are variable and mostly inconsistent due to a huge variability in the expression of the different markers on both healthy controls and AD cases.

### 4.8. Microglia Scoring System: A Proposal for Neuropathological Assessment

A validated method to grade microglial reactive states is yet to be conceived. In different pathological contexts, different methods of quantifying microglial activation have been developed, linked to those particular contexts and without a defined generalizability. These previous works investigated microglia in relation to diverse diseases (HIV infection, AD, multiple sclerosis) [22,181,216]. In our recent study on COVID-19 neuropathology, we observed that the microglial reaction plays a central role in COVID-19 cases but also in the forms of AD we studied independently of SARS-CoV-2 infection [121]. As observed also by other authors [217], SARS-CoV-2 induced an intense microglial reaction due to the cytokine storm and viral antigens penetrating into the CNS. Therefore, it may be considered a “perfect” activator of microglial reaction in the CNS, in a sort of “natural experiment on humans” due to the pandemic. Also, we believe that CD68 represents the paradigmatic immunohistochemical marker of microglial reaction, expressing better than anyone else the reactive (phagocytic) phenotype. In this frame, we coined a system to grade microglial reaction, which could also be used in other pathological contexts where a semi-quantitative evaluation of the microglial reaction is important. The scoring system we propose allows the examination of large areas of tissue; it is plain and easily reproducible. Microglial reaction is evaluated through the anti-CD68 antibody by an optical analysis (two-dimensional counting technique). The scoring system can be used for both grey and white matter and for both supratentorial and infratentorial structures. As for cortical grey matter, the hippocampus, and the white matter of hemispheres, we suggest examining three representative areas of 4.7 mm^2^ (corresponding to a 4× magnification field) across each slide (or section): upper-left corner, center, and lower-right corner. Considering the basal ganglia and the brainstem, we suggest examining an additional two fields: upper right and lower left, thus capturing all four corners of each slide plus the center area. A low magnification (4×) should be used to explore the area and count the reactive cells and nodules and higher magnifications (10–20×) to judge the cell morphology and microglial status. To evaluate microglial reactivity, we suggest not to count the total number of microglial cells positive for CD68 immunoreactivity but only those showing amoeboid morphology: with a larger body and stocky processes; indeed, a thin body is to be considered homeostatic even with CD68+. Moreover, it should be evaluated the presence of perivascular infiltrates and parenchymal clusters with three or more cells, namely microglial nodules [218]. This type of evaluation has its own complexity, which expresses the complexity of the cell population under examination and is not comparable to the simple evaluation, for example, of a proteinopathy (such as amyloid or pTAU), for which the automated evaluation of the area would be sufficient. This led us to choose the use of a manual scoring technique rather than automated quantification of the total antigen load, which would also include non-reactive microglial cells. The final scoring rises from a four-point semi-quantitatively evaluation scale (0–3), corresponding to none, mild, moderate, and severe microglial reaction: 0 = absence of both perivascular infiltrate and microglial nodules and <20 amoeboid cells/reactive microglial cells; 1 = presence of at least one perivascular infiltrate or 1 micronodule or >20 amoeboid cells/reactive microglial cells; 2 = presence of 2–4 microglial nodules; and 3 = presence of >4 microglial nodules. The scores of each area are then averaged to obtain a final value for each slide (or section). A 0 to 1 score may be considered as a condition of normal or homeostatic microglia. In Figure 5 there is a demonstrative representation of the grading system (Figure 5).

The semiquantitative scoring system just described and proposed, coined by Poloni & Medici [121], could be useful in neuropathological characterization, aiding in determining the intensity and topography of brain inflammation in diverse pathological conditions.

## 5. Concluding Remarks

Current evidence demonstrates that human microglial cells are a hugely varied and heterogeneous population. It should be also considered that most of the human microglial research has been conducted on Caucasians, with data from other ethnicities only now becoming available [219]. Microglial heterogeneity is crucial for neurodegeneration, although at the moment it was demonstrated mainly in neurodegenerative mice models [91]. These animal models can only partially clarify what happens in humans due to the fact that AD is a proper human disease which is complex and related to both genetic and environmental factors, with a trajectory of evolution that is different and peculiar for each patient. Just think of the association recently demonstrated between imaging markers of microglial reaction and behavioral symptoms in Alzheimer’s disease [220], which is certainly not transferable to mouse models. Thus, the role of microglia in human healthy aging and in AD presents multiple aspects, complex and interconnected. Although there are huge differences between humans and rodents, mouse models have been very useful to shed light on the microglial role in AD [221]. From our systematic review emerges that microglia have a fundamental role in removing pTAU and harmed synapses and in the phagocytosis and compaction of Aβ deposits. All these actions represent the protective aspects of microglia that are crucial to prevent neurodegeneration. This neuroprotective role may become less efficient with advancing age, primarily due to increased oxidative stress and mitochondrial dysfunction. Probably, the loss of efficiency of microglia and the accumulation of protein debris ends up determining a persistent mild inflammation. Therefore, in the brain areas where neurodegenerative phenomena are concentrated, possibly also associated with chronic hypoxia, a pathological context is created in which microglia lose their homeostatic role and become exhausted or dystrophic, otherwise they can become aggressive enhancing neurodegenerative phenomena and synapse loss. Thus, microglia may contribute to the progression of AD pathology in two ways: through functional exhaustion, with less efficiency in the removal of metabolic waste, or through neurotoxic phenomena due to an excess level of inflammation. Arguably, physiological aging and the maintenance of a healthy brain depends on establishing a balance between the actions and reactions of microglia. These lines of evidence suggest that microglia play a pivotal role in the pathogenesis of AD.

Many questions remain unanswered; for example: how do microglia move between their many states and how fluid are these states? Are they linked by a transcriptional continuum or by a hub-and-spoke connection pattern? Is it possible to have a therapeutic approach to favor the persistence of protective microglial phenotypes? Are the degenerative phenomena mainly due to the loss of function of the microglia or to excessive inflammation?

Therefore, in the future the combination of epigenetic, transcriptomic, metabolomic, and proteomic studies could be used to identify spatiotemporally different microglial subpopulations in mice and particularly in humans. According to recent evidence in the literature, we suggest that human microglia could be considered as a source of richness in terms of cell diversity. Investigating and better understanding this diversity would require the availability of well-characterized and fresh human brain tissue, obtained from well-studied brain donors [222]. In the future, this approach could open up new frontier scenarios regarding the role of microglia in aging and neurodegenerative diseases, also in anticipation of possible new therapies, and already now, disease-modifying therapies based on the cleaning of amyloids through monoclonal antibodies exploit the phagocytic action of microglia.

**Table 1 cells-12-02824-t001:** In this table, we report all studies that have analyzed microglia in neuropathological tissue brains from humans. The studies are reported in chronological order. For the various non-homeostatic microglial states, a number of biomarkers have been proposed. The existence of overlap between biomarkers for distinct states is in conflict with the binary ‘M1/M2’ classification. *Abbreviations: ADs:* Patients suffering from or dying with Alzheimer‘s disease; *M:* male subjects; *N/A:* value not available; *N/R:* value not reported; *LBD:* Lewy Bodies Dementia; *LATE:* Limbic-predominant Age-related TDP-43 Encephalopathy; *FTL:* Ferritin Light Chain; *DAM:* Disease Associated Microglia; *SASP:* Secretory Associated Senescence Pattern; *HAM:* Human Alzheimer’s Microglia; *LDAM:* Lipid-Droplet-Associated Microglia; *HS:* Hippocampal Sclerosis; *EOAD:* Early-Onset Alzheimer’s Disease; *LOAD:* Late-Onset Alzheimer’s Disease.

Study	Population	Age (Years Old; Mean ± SD)	Sex (M/F)	Microglial Phenotype	Cell-Surface Markers or Gene/Transcription Factors	Characteristics (e.g., Activity, Morphology, Interactions)	Method of Analysis	Interleukines/ Cytokines/ Chemokines (e.g., Stimulation, Production)
**DiPatre and Gelman (1997)** **[144]**	Younger Healthy Controls: 8	38	N/R	Primed	Ferritin	Young controls: sparse, not ramified Old controls: elongated bipolar microglia ADs: ramified processes, increased in number	Immunohistochemistry	N/A
	Older Healthy Controls: 9	73	N/R					
	ADs: 9	72	N/R					
**Sheng et al. (1998)** **[25]**	Healthy Controls (<60 y): 19	range: 1–57	7 M	Primed	N/A	Phagocytic, enlarged, extensive cytoplasm, rod shaped, ramified	Immunohistochemistry Western blot	IL-1α
	Healthy Aging (>60 y): 15	range: 61–93	12 M					
**Walker et al. (2001)** **[213]**	Healthy Controls: 5	N/R	N/R	DAM/WAM	CD11a, CD11b, CD11c, IL-2 R	Vacuolization after interaction with β-amyloid, activation and proliferation	Immunohistochemistry/RNA-seq	IL-1β, TNF- α, IL-6, IL-12β, IFN- γ
**Streit et al. (2004)** **[26]**	Young Healthy Controls: 1	38	1 M	Senescent	HLA-DR	Deramification, spheroid formation, gnarling, fragmentation of processes	Immunohistochemistry	N/A
	Old Healthy Controls: 1	68	1 M					
**Flanary et al. (2007)** **[66]**	Healthy Controls: 1	86	1 M	Senescent	Iba1, CD11b	Dystrophic	Immunohistochemistry	N/A
	ADs: 4	range: 82–89	3 M					
**Lopes et al. (2008)** **[36]**	Younger, non- demented individuals: 3	36.66 ± 2.08	4 M	Senescent	HLA-DR, Ferritin	HLA-DR+: ramified Ferritin+: dystrophic, deramified, fine processes tortuous and coiled	Immunohistochemistry Immunofluorescence	N/A
	Aged, non-demented and amyloid-free individuals: 7	79.86 ± 8.05	6 M					
	Aged, non-demented and high amyloid-β burden: 7	83.43 ± 5.19	5 M					
	ADs: 7	80.29 ± 11.64	4 M					
**Streit et al. (2009)** **[27]**	Healthy Controls: 4	range 22- 77	1 M	Senescent	Iba1	Dystrophic, cytoplasmic fragmentation, cytorrhexis	Immunohistochemistry	N/A
	Minimal tau pathology: 4	range 21–88	0 M					
	Maximal tau pathology without concurrent amyloid: 1	92	1 M					
	ADs: 4	range 62–85	2 M					
**Dhawan et al. (2012)** **[152]**	Healthy Controls: N/R	N/R	N/R	Primed	HLA-DR	Proliferation, microgliosis	Immunohistochemistry and Western blot	N/A
	ADs: N/R	N/R	N/R					
**Smith et al. (2013)** **[214]**	Autopsy brain tissue: N/A	N/A	N/A	Primed	HLA-DP,-DQ,-DR, CD45, PU.1, CX3CR1	Activation	Immunohistochemistry Western blot	N/A
	Post-mortem brain tissue: N/A	N/A	N/A					
**Griciuc et al. (2013)** **[223]**	Healthy Controls: 15	N/R	N/R	Primed	Iba1, CD33	Activation	Western blot qT-PCR	N/A
	ADs: 25	N/R	N/R					
**Bachstetter et al. (2015)** **[224]**	Healthy Controls: 9	86	6 M	Primed—Senescent	Iba 1, CD68	Ramified, hypertrophic, dystrophic, rod shaped, amoeboid	Immunohistochemistry	N/A
	HS of aging: 6	87	3 M					
	ADs: 7	77	4 M					
	ADs + HSs: 4	91	0 M					
	MCIs: 414	71.82 (7.45)	244 M					
	ADs: 73	74.17 (8.37)	38 M					
**Zeineh et al. (2015)** **[209]**	Healthy Controls: 8	72.8	3 M	Primed—M1	CD163	Reactive microgliosis	Immunoblotting	N/A
	ADs: 5	85.8	2 M					
**Tischer et al. (2016)** **[225]**	Healthy Controls: 5	71.9 ± 6.8	11 M	Senescent	Iba1, CD68, MHCII	Dystrophic Younger controls: ramified branches Older controls: loss of branches and ramification ADs: fragmented, spheroid, shortening of branches	Immunohistochemistry Immunofluorescence	N/A
	Prodromal ADs: 11	80.0 ± 7.7						
	Progressive ADs: 10	26–30/78–83						
**Satoh et al. (2016)** **[226]**	Sporadic ADs: 10	70 ± 8	5 M	Primed	TMEM119, TREM2, Iba1, CD68	Ramified and amoeboid morphologies	Immunohistochemistry Western blot Real-Time PCR	N/A
	Non-ADs: 11 (Healthy Controls: 4)	75 ± 8	6 M					
**Hendrickx et al. (2017)** **[212]**	Healthy Controls: 6	64.8	2 M	Primed	Iba1, CD68, HLA-DR	Ramified microglia, rounded amoeboid microglia, foamy macrophages	Immunohistochemistry	N/A
	ADs: 4	72.3	2 M					
**Raj et al. (2017)** **[227]**	Healthy Controls: 20	range: 30–85	N/R	Primed	CD68, Iba1, HLA-DR	Activation, proliferation	Immunohistochemistry	N/A
	ADs (EOAD + LOAD): 12	range: 20–80	N/R					
**Bachstetter et al. (2017)** **[81]**	Healthy Controls: 118	range: 20–70	107 M	M2	Iba1	Rod-shaped microglia	Immunohistochemistry	N/A
	ADs: 50	range: 70–89						
**Sims et al. (2017)** **[228]**	Healthy Controls + ADs: 508	76.3 (Healthy Controls) 75.9 (ADs)	N/R N/R	DAM/WAM	TREM2, ABI3, PLCG2	N/A	Genotyping	N/A
**Davies et al. (2017)** **[68]**	Younger Healthy Controls: 3	55 ± 4	2 M	Senescent	Iba1	Dystrophic, ramified, deramified, discontinuous, punctate	Immunohistochemistry Immunofluorescence	N/A
	Older Healthy Controls: 5	82 ± 10	2 M					
	ADs: 7	84 ± 11	3 M					
**Krasemann et al. (2017)** **[87]**	ADs (TREM2 variants): 5	80.6	3 M	M0 homeostatic microglia	Iba1, TREM2, TMEM119, P2RY12, APOE	Homeostatic microglia; Proliferation and clustering around β-amyloid plaques	Immunohistochemistry Immunofluorescence	N/A
	ADs (TREM2 wild-type): 6	81	4 M					
**Olah et al. (2018)** **[153]**	Healthy Controls: 4	N/R	N/R	DAM/WAM	CD11, CD45, P2RY12, TMEM119, TREM2, GPR34, CX3CR1	Activation, dystrophic	RNA-seq	N/A
**Kaneshwaran et al. (2019)** **[229]**	Healthy Controls: 420	86.6–93.3	233 M	Primed	HLA-DP, DQ, and DR	Dystrophic **Stage I**: thin ramified processes **Stage II**: plump cytoplasm and thicker processes **Stage III**: appearance of macrophages	Immunohistochemistry	N/A
	ADs: 265	89.4 (86.3–92.9)						
**Mukherhjee et al. (2019)** **[230]**	ADs + Healthy Controls: 637	range: 18–106	319 M	DAM/WAM	FCER1G, ITGB2/CD18, MYO1F, PTPRC/CD45, TYROBP/DAP12	N/A	WGCNA meta-analysis RNA seq	N/A
**Parhizkar et al. (2019)** **[231]**	Non-AD Dementia: 3	80.7	8 M	Primed—Homeostatic	Iba1, TREM2	Clustering behavior around β-amyloid plaques and reduced clustering in patients with TREM2 loss of function variants	Immunohistochemistry Immunoblotting/Genotyping	N/A
	ADs: 7	72.4	1 M					
**Barroeta-Espar et al. (2019)** **[181]**	Healthy Controls: 28	86.1 ± 10.1/86.6 ± 10.2	18 M	Primed	CD68	Activation, proliferation	Immunohistochemistry	Associated with resilience to AD: upregulation of IL-1β, IL-6, IL-13, IL-4, IL-6, IL-10, IP-10, PDGF-bb, FGF; GM-CSF, IL-17, IL-7 Associated with AD: upregulation of IL-1α and TNF-α, IL-5, IL-8, IL-12p70, MCP-1, MIP-1 α, eotaxin, IL-1ra
	Non-Demented (Int-med risk): 33	83.1 ± 11.6	18 M					
	ADs: 29	82.9 ± 11.6	18 M					
**Felsky et al. (2019)** **[84]**	ADs: 71	78 ± 8.7	N/R	Primed DAM	N/A	Activation, dystrophic **Stage I**: thin ramified processes **Stage II**: plump cytoplasm and thicker processes **Stage III**: appearance of macrophages	Immunohistochemistry	N/A
	Postmortem ADs: 90	N/R	N/R					
**Bonham et al. (2019)** **[215]**	Healthy Controls: 6	N/R	N/R	Senescent	TMEM119	Dystrophic	Gene expression mapping	N/A
	ADs: 584	N/R	N/R					
**Study**	**Population**	**Age** **(Years Old; Mean ± SD)**	**Sex** **(M/F)**	**Microglial** **Phenotype**	**Cell Surface Antigens/Biomarkers** **(e.g., Transcription Factors)**	**Morphological Characteristics**	**Methods**	**Interleukines/** **Cytokines/** **Chemokines** **(e.g., Stimulation, Production)**
**Li et al. (2020)** **[232]**	Healthy Controls: 10	83 ± 2	6 M	Senescent	Iba1, TPSO, TREM2, MPO, BIN 1	Dystrophic	Immunohistochemistry Genotyping	N/A
	ADs: 27	82 ± 2	13 M					
**Walker et al. (2020)** **[166]**	Low plaque non-demented: 12	85.9 ± 8.9	6 M	Primed—Senescent	CD68 P2RY12	Ramified, dystrophic, “tufted”	Immunohistochemistry Western blot RNA expressing profile	Stimulation with: IL-4, IL-6, IFN- γ
	High plaque non-demented: 12	88 ± 8	4 M					
**Srinivasan et al. (2020)** **[106]**	Healthy Controls: 21	79	10 M	HAM	CD11b	Activation, proliferation	RNA-seq	N/A
	ADs: 21	80	13 M					
**Molina-Martinez et al. (2020)** **[49]**	Healthy Controls: 8	54 ± 2/65 ± 1.8	11 M	Primed	N/A	Activation, proliferation	Genotyping	N/A
	ADs: 17	51 ± 1.8	7 M					
**Olah et al. (2020)** **[182]**	Healthy Controls: 11	N/A	N/A	DAM/WAM	Iba1, ISG15+, CD83+, and PCNA+, CD74, AIF2, APOE. TREM2	Ramified, ameboid	Immunohistochemistry RNA-seq	IL-10, IL-4, IL-13, IFN- γ
	MCI: 4	95	1 M					
	ADs: 10	91	2 M					
**Friedberg et al. (2020)** **[183]**	ADs APOE ε4 negative: 34	87.9 ± 0.843	15 M	Primed—M1	Iba1, CD68	Activation	Immunohistochemistry	IL-1α, IL-4, IL-13
	ADs APOE ε4 positive: 21	85.9 ± 1.63	10 M					
**Fadul et al. (2020)** **[233]**	Healthy Controls: 97	85.6	N/R	DAM/WAM	CD 68, MHCII, NDRG2 (Astrocyte marker)	N/A	Immunohistochemistry	N/A
**Marschallinger et al. (2020)** **[107]**	Young Healthy Controls = 3	<30	N/R	LDAM	BODIPY, Iba1, CD68, TMEM119	Pro-inflammatory phenotype Phagocytic deficits	Immunohistochemistry RNA-seq	IL-10, CCL3, CCL4, IL-6, CCL5, TNF-α, IL-1β, IL-1α, CXCL1, CXCL10
	Older Healthy Controls = 5	>60	N/R					
**Pascoal et al. (2021)** **[145]**	Healthy Controls: 86	23 ± 2.4 (young)/ 72 ± 5.5 (older)	22 M	Primed	TREM2	Proliferation	Multiplex immunoassay analysis	TRAIL, CXCL1, CX3CL1, TGF- α, CCL3, CCL23, IL-8
	ADs: 16	70 ± 7.7	6 M					
	MCIs: 28	73 ± 8.6	17 M					
**March-Diaz et al. (2021)** **[165]**	Healthy Controls: N/R	49.5 ± 5.9	N/R	Primed DAM	Iba1	Reduced clustering around β-amyloid plaques	Western blot	N/A
	ADs: N/R	78 ± 8.5/78.3 ± 14.0/79 ± 10.0	N/R					
	ADs: N/R	N/R	N/R					
**Kloske et al. (2021)** **[234]**	Healthy Controls APOE ε3/3: 9	81 (73–90)	4 M	Primed	P2RY12	Activation	Immunohistochemistry	N/A
	ADs APOE ε3/3: 9	81 (72–87)	6 M					
	ADs APOE ε3/3: 10	85 (75–95)	2 M					
**Hu et al. (2021)** **[57]**	Healthy Controls: 7	74.28	4 M	Senescent/SASP	Iba1, PAI1, P19, P16, P21, CASPASE-8 (CASP8)	Distrophyc	Immunohistochemistry	IL-1β, IL-6
	ADs: 7	70.57	5 M					
**Cohn et al. (2021)** **[184]**	Low-Mild ADs: 4	85	3 M	Primed DAM	CD11b, TMEM119, P2RY12, TREM2, FTH1	Activation, proliferation	Immunoblotting/Transcriptomics	TNF-α
	Moderate-Severe ADs: 4	91	1 M					
**Shahidehpour et al. (2021)** **[67]**	Healthy Controls: 34	range: 65–93	13 M	Senescent	Iba1, FTL	Hypertrophic, ramified, dystrophic	Immunohistochemistry	N/A
	ADs: 8	range: 65–85	5 M					
	LATEs: 9	range: 65–93	4 M					
	LBDs: 11	range: 65–97	9 M					
**Jiang et al. (2022)** **[235]**	Healthy Controls: 11	89.8	4 M	Senescent	Iba1	Dystrophic activation, proliferation	Immunohistochemistry	sST2/IL-33
	ADs: 102	80	47 M					
**Xie et al. (2022)** **[236]**	Healthy Controls: 4	Mean age cases: 84.25	3 M	Primed	Iba1, CatE	Activation, proliferation	Immunoblotting	N/A
	LOADs: 4	Mean age controls: 85	2 M					
**An et al. (2022)** **[58]**	Human microglial cells HMC3 ATCC (#CRL0314) Human Brain Bank derived	N/A	N/A	Senescent—SASP	β-galactosidase (SA-β-gal), SIRT1/NRF2 pathway	Dystrophic	Western blot Real-Time PCR	TNF-α, IL-1β, IL-6
**Muñoz-Castro et al. (2022)** **[237]**	Healthy Controls: 7	86.0 ± 2.5	4 M	DAM/WAM	Iba1, CD68, ferritin, MHCII, TMEM119, TSPO	Homeostatic, intermediate, reactive	Immunohistochemistry Immunofluorescence Machine learning	N/A
	ADs: 7	76.7 ± 11.2	3 M					
**Neumann et al. (2023)** **[64]**	Healthy Controls: 14	80.4	6 M	Senescent	Iba1, y-H2AX, 8-OHdG, HO-1, lamin B1, ferritin	Dystrophic, ramified	Immunohistochemistry	N/A
**Aghaizu et al. (2023)** **[155]**	Healthy Controls: 5	81.6	2 M	Primed	Iba1	Proliferation and clustering around β-amyloid plaques	Immunohistochemistry	N/A
	ADs: 6	64.5	2 M					
	non-AD Dementia: 2	92	0 M					

**Table 2 cells-12-02824-t002:** Main antibody markers * used to visualize human microglia in Healthy Aging and Alzheimer’s disease (AD).

Marker	Specificity	Labeled States	Staining Patterns	Main Applications	Reference
**Iba1**	macrophages including microglia	homeostatic conditions and disease associated	visualization of microglial cell body and processes, distal extremities. diffuses throughout the cytoplasm	categorization into morphological states, microglial density distribution	Aghaizu et al. (2023) [155] March-Diaz et al. (2021) [165] Tischer et al. (2016) [225] Bachstetter et al. (2017) [81] Bachstetter et al. (2015) [224] Raj et al. (2017) [227] Jiang et al. (2022) [235] Parhizkar et al. (2019) [231] Streit et al. (2009) [27] Shahidehpour et al. (2021) [67] Hu et al. (2021) [57] Flanary et al. (2007) [66] Li et al. (2020) [232] Griciuc et al. (2013) [223] Xie et al. (2022) [236] Olah et al. (2020) [182] Friedberg et al. (2020) [183] Davies et al. (2017) [68] Muñoz-Castro et al. (2022) [237] Marschallinger et al. (2020) [107] Zhao et al. (2022) [238]
**CD11b/c**	macrophages including microglia	homeostatic conditions and disease associated	low basal expression in adult microglia, Staining is mainly restricted to the plasma membrane	categorization into morphological states, microglial density distribution, morphology ultrastructural studies of subsets downregulating IBA1	Cohn et al. (2021) [184] Walker et al. (2001) [213] Srinivasan et al. (2020) [106] Flanary et al. (2007) [66] Olah et al. (2018) [153]
**P2RY12**	microglia specific, state dependent	homeostatic marker, strongly downregulated in disease associated and reactive states	staining can localize the plasma membrane or diffuse throughout the cytoplasm	analysis of microglial density, distribution, and morphology ultrastructural studies	Cohn et al. (2021) [184] Walker et al. (2020) [166] Kloske et al. (2021) [234] Olah et al. (2018) [153]
**TMEM119**	largely microglia specific, state dependent	homeostatic conditions and disease associated	microglial cell bodies and staining of their processes	categorization into morphological states, microglial density distribution, morphology ultrastructural studies in combination with IBA1^+^-CD68^+^ cells (ramified and amoeboid morphologies)	Satoh et al. (2016) [226] Cohn et al. (2021) [184] Bonham et al. (2019) [215] Olah et al. (2018) [153]
**TREM2**	macrophages including microglia, state dependent	aging and disease conditions (e.g., amyloid plaques in AD pathology)	visualization of microglial cell body and processes, distal extremities. diffuses throughout the cytoplasm	categorization into morphological states, microglial density distribution, morphology ultrastructural studies of subsets downregulating IBA1^+^ cells	Fahrenhold et al. (2018) [105] Krasemann et al. (2017) [87] Satoh et al. (2016) [226] Parhizkar et al. (2019) [231] Li et al. (2020) [232] Olah et al. (2020) [182] Sims et al. (2017) [228] Cohn et al. (2021) [184] Parhizkar et al. (2019) [231] Pascoal et al. (2021) [145]
**HLA-DR**	macrophages including microglia	homeostatic conditions and disease associated	visualization of microglial cell body and processes (ramification and deramification), distal extremities. diffuses throughout the cytoplasm	categorization into morphological states, microglial density distribution, morphology ultrastructural studies	Dhawan et al. (2012) [152] Raj et al. (2017) [227] Smith et al. (2013) [214] Lopes et al. (2008) [36] Kaneshwaran et al. (2019) [229] Streit et al. (2004) [26]
**CD68**	macrophages including microglia predominantly expressed by lysosomes	reactive states, disease associated	because lysosomes are mostly found near the nucleus in ramified and amoeboid microglia, the characteristic extrusions cannot be seen with CD68 labeling.	categorization into morphological states, microglial density distribution	Hendrickx et al. (2017) [212]
**MHCII**	macrophages including microglia	homeostatic conditions and disease associated	visualization of microglial cell body and processes, distal extremities diffuses throughout the cytoplasm	categorization into morphological states	Muñoz-Castro et al. (2022) [237] Tischer et al. (2016) [225] Fadul et al. (2020) [233]
**CX3CR1**	macrophages including microglia	homeostatic conditions and disease associated	visualization of microglial cell body and processes	microglial density, distribution, and categorization into morphological states	Keren-Shaul et al. (2017) [91] Krasemann et al. (2017) [87] Olah et al. (2018) [153] Smith et al. (2013) [214]

* Other proteins expressed by human microglia but whose specificity is not confirmed include APOE, CLEC7A, ITGAX, and LPL.

## Figures and Tables

**Figure 1 cells-12-02824-f001:**
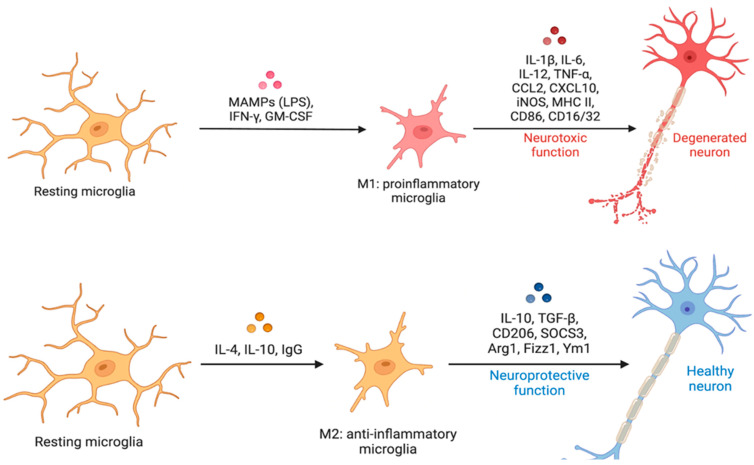
Microglia Classical Activation and Microglia Alternative Activation. (**On the top**): «M1» or classical activation is thought to be pro-inflammatory and neurotoxic and is closely tied to the idea of “reactive” microglia. (**On the bottom**): «M2» or alternative activation is thought to be anti-inflammatory and neuroprotective. When activated by LPS, IFN-γ, or GM-CSF, microglia develop an M1 pro-inflammatory phenotype, which leads to neurotoxicity by secreting various pro-inflammatory chemicals. When stimulated by IL-1, IgG, or IL-10, microglia enter an anti-inflammatory M2 state, resulting in neuroprotection via the release of a range of chemicals. Abbreviations: Arg1, arginase 1; CCL, chemokine (C-C motif) ligand; CD, cluster of differentiation; CSF1R, colony-stimulating factor-1 receptor; CXCL, chemokine (C-X-C motif) ligand; Fizz1, found in inflammatory zone; IL, interleukin; GM-CSF, granulocyte-macrophage colony-stimulating factor; IFN-γ, interferon-γ; iNOS, inducible nitric oxide synthase; LPS, lipopolysaccharide; MAMPs, microbe-associated molecular patterns; MHC-II, major histocompatibility complex II; TNF-α, tumor necrosis factor-α; Ym1, chitinase-like protein.

**Figure 2 cells-12-02824-f002:**
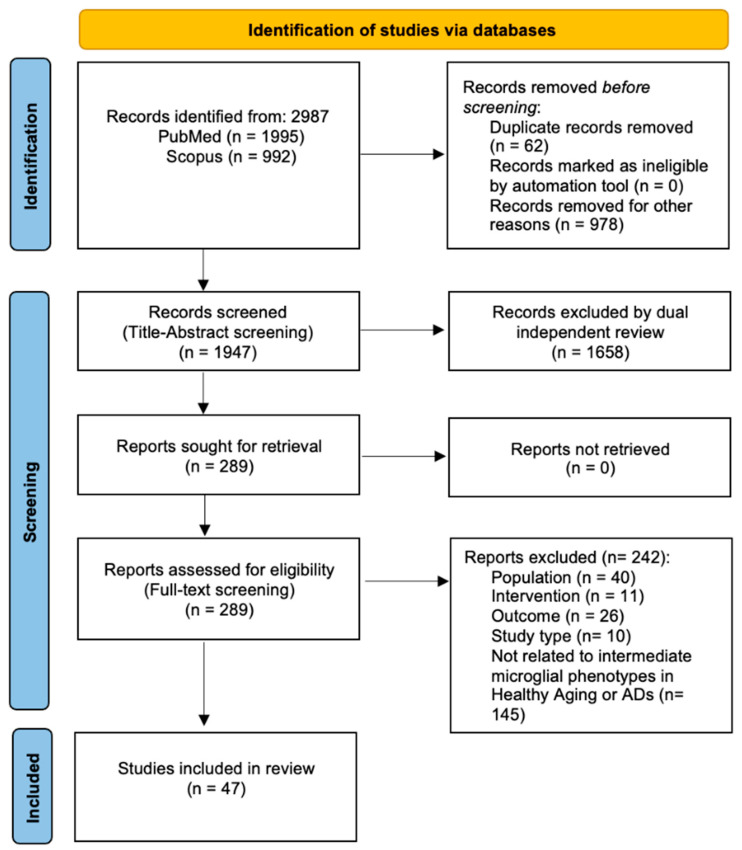
The study selection process using the PRISMA flow diagram [55].

**Figure 3 cells-12-02824-f003:**
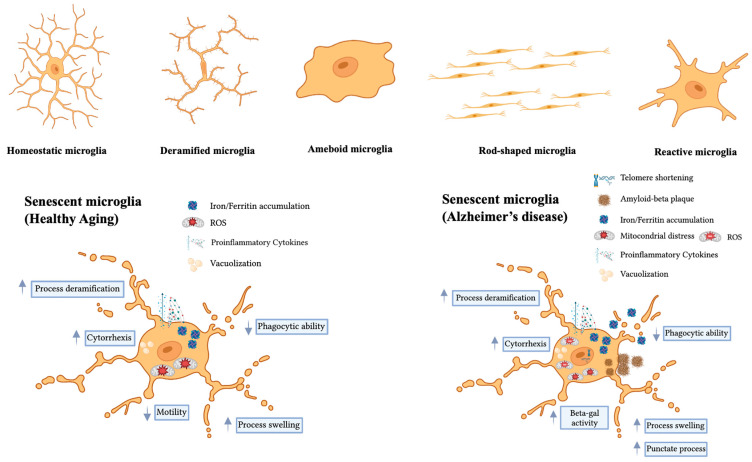
Microglial Senescence in Healthy Aging and Alzheimer’s Disease. (**On the top**)*:* Different morphological patterns of microglial stages from homeostatic state to reactive microglia. (**On the bottom** (**left**)): Microglial aging characteristics. Microglia lose process ramification, develop process defects, and exhibit cytoplasmic fragmentation as they mature with vacuolization. They have higher iron storage, ferritin expression, and higher ROS production. Higher production of neurotoxic chemicals and proinflammatory cytokines and decreased ability to phagocytize debris and toxic protein complexes. (**On the bottom** (**right**)): AD’s microglial characteristics. Microglia lose process ramification, develop process defects, and exhibit cytoplasmic fragmentation as they mature with vacuolization. They have higher iron storage, ferritin expression, higher ROS production, and higher production of neurotoxic chemicals and proinflammatory cytokines. Increased production of neurotoxic chemicals and decreased ability to phagocytize debris and toxic protein complexes. Moreover, AD’s microglia may have high burden of Amyloid-beta plaque, high beta-galactosidase activity, and telomere shortening. Microglial state alterations are linked to changes in shape, gene expression, and behavior. Microglia associated with disease or reactive microglia have an amoeboid morphology, retracted processes, and enhanced phagocytosis. Dystrophic or senescent microglia show cytorrhexis as well as a reduction in phagocytosis and motility, while in a surveillance state, microglia can expand and retract their processes, creating short interactions with synaptic sites. Abbreviations: ROS (Reactive Oxidative Species); Beta-gal (Beta-galactosidase).

**Figure 4 cells-12-02824-f004:**
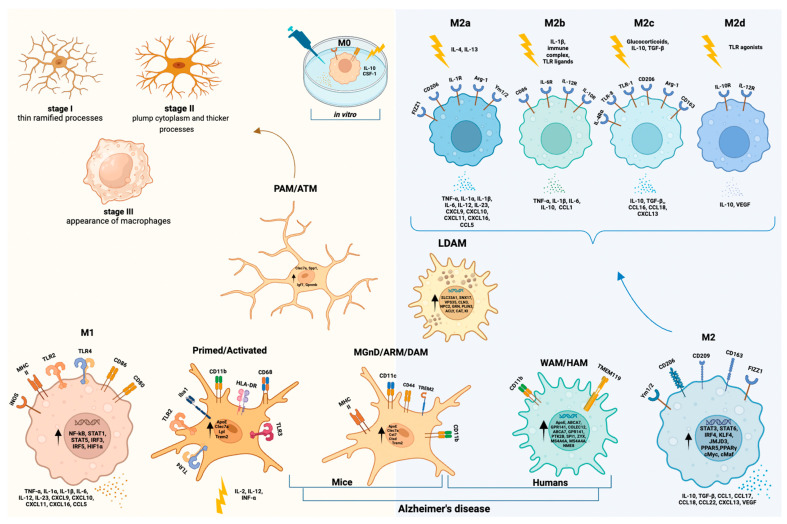
Representation of microglial states and functions. This scheme represents the extreme complexity and variability of microglial state scenarios in humans and mouse models, resulting from our systematic review results. (**On the bottom**): Microglia have been traditionally framed into dichotomic categories, but our current integration of epigenetic, transcriptomic, metabolomic, and proteomic data favors a multidimensional integration of coexisting states. Many microglial transcriptional signatures have been identified, including PAM, ATM; MGnD/ARM/DAM in mice models of AD; and WAM/HAM/LDAM in human aging and AD. The key genes upregulated (black arrow) in each signature are indicated. Regarding different reactive microglial states, numerous types of markers have been suggested. The existence of overlap between markers for distinct states runs counter to the binary ‘M1/M2’ paradigm. (**On the top**): A related term is “M0” microglia, which defines their state when cultured in the presence of IL-10 and CSF1 to imitate their in vivo counterparts. (**On the top** (**left**)): PAM/ATM may share some features with the core DAM/WAM signature. However, in early post-natal mice models, this phenotype can differentiate into three different microglial stages: (I) thin ramified processes; (II) plump cytoplasm and thicker processes; (III) appearance of macrophages. (**On the top** (**right**)): A shift towards M2a-d subclassification has been proposed. In this classification, the traditional alternative activation phenotype (neuroprotective), designated as M2a, is characterized by upregulation of Arg1 and scavenger receptors, along with inhibition of NF-κB isoforms. M2b identifies microglia involved in immunoregulation, while M2c and M2d are involved in tissue remodeling. Abbreviations: Arg1, arginase 1; CCL, chemokine (C-C motif) ligand; CD, cluster of differentiation; CSF-1, colony-stimulating factor-1; CXCL, chemokine (C-X-C motif) ligand; Fizz1, found in inflammatory zone; IL, interleukin; GM-CSF, granulocyte-macrophage colony-stimulating factor; IFN-γ, interferon-γ; iNOS, inducible nitric oxide synthase; MHC-II, major histocompatibility complex II; TNF-α, tumor necrosis factor-α; Ym1, chitinase-like protein; TMEM119, Transmembrane Protein 119; CLEC7a, C-type lectin domain containing 7°; PLIN, Perilipins proteins; TLR, Tool-Like Receptor; Iba1, Ionized calcium binding adaptor molecule 1; PAM, proliferative-associated microglia; ATM, axon tract-associated microglia; DAM, disease-associated microglia, MGnD, microglial neurodegenerative phenotype; ARM, activated response microglia; WAM, white matter-associated microglia; HAM, human AD microglia; LDAM, lipid droplet accumulating microglia.

**Figure 5 cells-12-02824-f005:**
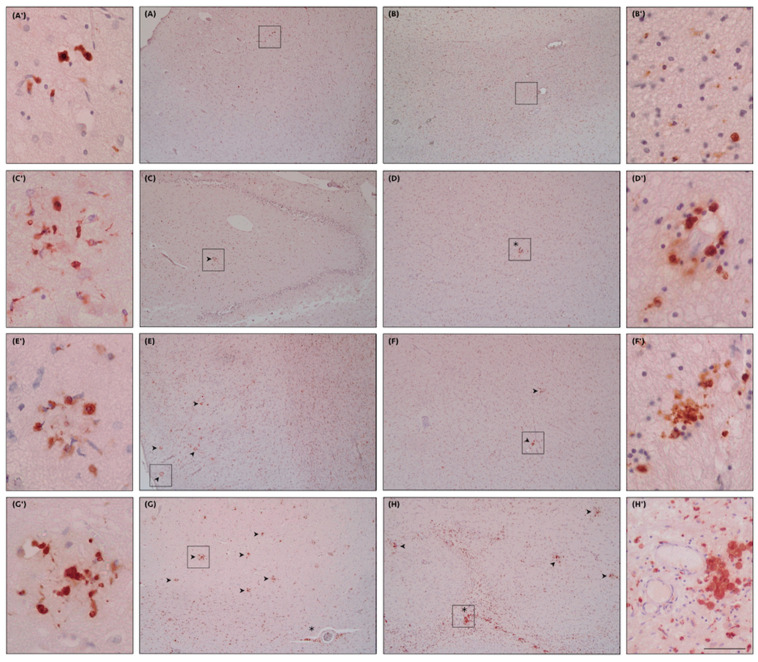
Microglial Semi-Quantitative Grading System. CD-68 Immunohistochemistry is performed for the analysis. The second and third columns show the assessment of microglial activation through the proposed manual semi-quantitative grading system in the grey and white matter, respectively, at low microscopical magnification (4×). The first and fourth columns show higher microscopical magnification (**A’**–**G’** at 40× and **H’** at 20×) of a detail of the respective low magnification (4×) images (black square). Microglial activation is rated on a 4-point scale (0–3): 0 = absence of both perivascular infiltrate and microglial nodules and <20 amoeboid/reactive microglial cells (**A**,**B**); 1 = presence of at least one perivascular infiltrate or 1 micronodule or >20 amoeboid cells/reactive microglial cells (**C**,**D**); 2 = presence of 2–4 microglial nodules (**E**,**F**); and 3 = presence of >4 microglial nodules (**G**,**H**). This proposed scoring system uses a low magnification to explore the area and higher magnifications to evaluate the cell morphology and microglial activation status. Regarding the morphology, the image shows “rod-shaped” microglia (**A’**), perivascular infiltrates (**D**,**H**; asterisks), and microglial nodules (**C**,**E**–**H**; arrowhead). Scale bars: 515 µm (**A**–**H**); 31 μm (**A’**–**G’**); 105 μm (**H’**).

## Data Availability

The data details used to reproduce the vast majority of the results are provided in Zenodo (https://doi.org/10.5281/zenodo.10154404).

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
