# Peer review of "Microglial Senescence and Activation in Healthy Aging and Alzheimer’s Disease: Systematic Review and Neuropathological Scoring"

_cells, 2023, doi:10.3390/cells12242824_

Round 1

Reviewer 1 Report

Comments and Suggestions for Authors

The review article entitled "Microglial Senescence and Activation in Healthy Aging and Alzheimer’s Disease: Systematic Review and Neuropathological Scoring" discusses the microglial transcriptomic, phenotypic and neuropathological profiles in healthy aging and AD. Overall the article is timely, descriptive, very well written and informative. There are few minor concerns the authors need to address:

Minor Comments:

1. The authors are suggested to add a section, how change in microglial phenotype can affect the neigboring cells.

2. Please check for typos.

Comments on the Quality of English Language

Minor revisions are required.

Author Response

Minor Comments:

  1. The authors are suggested to add a section, how change in microglial phenotype can affect the neigboring cells. Answer: We agree that the actions of microglia are best understood by adding details about how microglia cooperate with other cells; we did this in several places in the manuscript, including adding a sentence in the abstract (partially rearranged to stay within 200 words). The changes can be found in the text of the introduction and discussion (section 4.4), highlighted in red, and accompanied by relevant quotes.
  2. Please check for typos. We did it carefully

We thank you very much for the time you dedicated to the critical reading of our article, giving us valuable advice that have certainly improved the manuscript.

 Dr T.E. Poloni and coauthors

Reviewer 2 Report

Comments and Suggestions for Authors

The article titled “Microglial senescence and activation in healthy aging and Alzheimer´s disease: Systematic review and neuropathological scoring” by Malvaso et al. is a well-written, comprehensive and updated review on the different subtypes and states of microglia as well as on their roles in brain homeostasis, neuro-inflammation and neuroprotection.

I have two requests for the authors to consider:

1.    Relevant studies using human induced pluripotent stem cells (hiPSCs) to generate microglia as a model to study Alzheimer´s disease should be discussed in this review.

2.    Fig. 5: Please include high magnification insets to better visualize microglial morphology.

Author Response

Minor Comments:

I have two requests for the authors to consider:

  1. Relevant studies using human induced pluripotent stem cells (hiPSCs) to generate microglia as a model to study Alzheimer´s disease should be discussed in this review. Answer: We agree that the study through cellular models is a hot topic to deepen the understanding of human microglia, especially to generate research hypotheses that will then be confirmed through tissue analysis. We have added a part on these aspects in the section 4.4 of the discussion (highlighted in red), accompanied by relevant quotes.
  2. Fig. 5: Please include high magnification insets to better visualize microglial morphology. Answer: We did it and now the image is very good, please see it.

We thank you very much for the time you dedicated to the critical reading of our article, giving us valuable advice that certainly have improved the manuscript.

 Dr T.E. Poloni and coauthors

Reviewer 3 Report

Comments and Suggestions for Authors

The authors emphasize the microglial signatures in homeostatic brain aging and Alzheimer's disease (AD). A systematic literature search of all published articles about microglial senescence in healthy human aging and Alzheimer's disease was performed. The authors provide published data on morphological, functional and transcriptomic heterogeneity of microglia. But the authors deviated from their primary target, aging and Alzheimer's disease, and wrote about viral infections such as COVID-19 and other conditions that are not their primary target. All these deviations from the main topic weigh down the text and should be removed. The authors should maintain the basic topics to ensure the clarity of the manuscript.

The citation style is not appropriate throughout the manuscript.

Author's name without year and serial number from a reference list is not appropriate citation style. Please correct this throughout the manuscript.

e.g. Please insert a reference after the statements:

lines 88-89, Angelova et al. reviewed the differences between these two terms.

also lines 271-274
lines 392-396 insert citation for statement
lines 528-530 insert citation for statement lines 535-536 insert citation for statement

etc...

Whole subsection 4.3 Homeostatic microglia: responses to external/internal perturbations and aging is too long, unsystematized and lacks appropriate citations.
e.g. : line 621-624 insert a citation for the statement
 Line 635-637, Line 640-641, Line 648-650 insert citation for the statements
Line 670-675 there are no citations for the statement
Line 708-710
Line 716-722 there are no citations for the statement
Line 724-732 there are no citations for the statement

e.g. lines 770-772, 778-780, 782-784, 788-790 etc. insert a citation for the statement

Please check the entire manuscript. The citations are myssing to the end of the manuscript.

In Table 1 and Table 2, please include the number of the citation along with the name and year from the bibliography to make the reference easier to find in the bibliography.

If you are using the term for the first time, write the name in full, followed by the abbreviation in brackets.

line 421, line 431, line 438,line 486 Please insert what the abbreviations PAM/ATM, DAM/WAM,DAM/MGnD/ARMWAM/HAM/LDAM, TYROBP mean ... And etc. until the end of the manuscript there is no meaning of the abbreviations.

Author Response

Minor Comments:

Answer: We have considerably reduced the parts on COVID-19 and eliminated several general and generic considerations in the section 4.3 of the discussion; however, we have left some aspects regarding the response of human cerebral microglia to environmental insults and stimuli, due to the importance that these may have in the trajectory of the human brain aging, in which microglia dysfunctions may lead to the onset of AD pathology. We have also corrected all the small errors that you kindly reported to us, and arranged the manuscript accordingly.

We thank you very much for the time you dedicated to the critical reading of our article, giving us valuable advice that have certainly improved the manuscript.

                                                                                    Dr T.E. Poloni and coauthors